# Non-Parallel Text Style Transfer with Self-Parallel Supervision

**Ruibo Liu♠, Chongyang Gao♣, Chenyan Jia♥, Guangxuan Xu♦, Soroush Vosoughi♠**

♠Dartmouth College, ♣Northwestern University, ♥University of Texas, Austin
♦University of California, Los Angeles

♠`{ruibo.liu.gr, soroush.vosoughi}@dartmouth.edu`
♣`cygao@u.northwestern.edu`
♥`chenyanjia@utexas.edu`
♦`gxu21@cs.ucla.edu`

## Abstract

The performance of existing text style transfer models is severely limited by the non-parallel datasets on which the models are trained. In non-parallel datasets, no direct mapping exists between sentences of the source and target style; the style transfer models thus only receive weak supervision of the target sentences during training, which often leads the model to discard too much style-independent information, or utterly fail to transfer the style.

In this work, we propose LaMer, a novel text style transfer framework based on large-scale language models. LaMer first mines the roughly parallel expressions in the non-parallel datasets with scene graphs, and then employs MLE training, followed by imitation learning refinement, to leverage the intrinsic parallelism within the data. On two benchmark tasks (sentiment & formality transfer) and a newly proposed challenging task (political stance transfer), our model achieves qualitative advances in transfer accuracy, content preservation, and fluency. Further empirical and human evaluations demonstrate that our model not only makes training more efficient, but also generates more readable and diverse expressions than previous models.

## 1 Introduction

Text style transfer (TST) models learn how to transfer the style of text from source to target while preserving the style-independent content (John et al., 2019; Fu et al., 2018). Existing TST methods perform well when transferring simple styles, such as sentiment; however, they tend to do a poor job on more abstract and subtle styles, such as formality and political stance (Lee et al., 2021; Fu et al., 2019b).

The lack of parallel datasets is one of the main bottlenecks for text style transfer tasks. Parallel datasets contain pairwise sentences that only differ in specific styles, while non-parallel datasets typically contain a large corpora of sentences with different styles, without direct one-to-one mapping between sentences (Fu et al., 2019a; Krishna et al., 2020). In real-world settings, it is much easier to construct non-parallel datasets by collecting large quantities of non-parallel sentences with different styles; conversely, carefully-aligned parallel TST datasets are much more costly to construct, usually requiring human annotation, and are thus scarce.

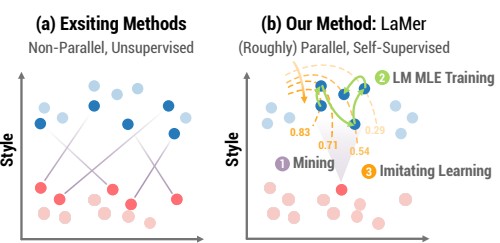

Figure 1: Red and blue circles represent the source and target texts respectively. (a) Existing methods crucially ignore the inherit parallelism within the data. (b) Our method first mines (roughly) parallel expressions, then learns how to transfer style with the self-supervision from the parallel expressions.

The lack of parallel datasets for style transfer tasks prompts researchers to develop unsupervised TST methods trained on non-parallel datasets (Dai et al., 2019; Luo et al., 2019; Shen et al., 2017). Current methods either rely on surface-level split of style- and content-bearing tokens (Li et al., 2018a; Castro et al., 2017), or separate content and style vectors in latent space (John et al., 2019; Liu et al., 2020a; Wang et al., 2019) — once style and content have been separated, they then swap styles leaving the content unchanged. However, surface-level split is not reliable when the style cannot be extracted at the token level (Sudhakar et al., 2019); latent-space disentanglement could also fail due to a lack of pairwise supervision and results in discarding too much style-independent content (Reif et al., 2021). However, these methods are less than ideal. As demonstrated in Figure 1 (a), they tend to learn the mapping between source and target styles on the original randomly mapped sentence pairs, but fail to take into account the inherent parallelism within the data (Krishna et al., 2020).

This paper presents LaMer, a novel self-supervised text style transfer framework built upon recent text-to-text language models (LMs). As illustrated in Figure 1 (b), rather than accepting the random mapping in the given non-parallel dataset, we first align sentences from source and target styles based on their scene graphs, and then run conditional MLE training on LM to maximize the probability of generating (roughly) parallel style sentences. As a final refinement step, we deploy reinforced imitation learning with contrastive loss on the aligned data to further reward the best transfer demonstration and penalize the others. In comparison with existing methods, LaMer fully exploits the intrinsic parallelism within non-parallel data and innovatively applies the transfer learning power of large-scale LMs to boost text-style transfer performance.

Our contributions can be outlined as follows: *First*, We present a language model based text style transfer framework, LAMER, that achieves significant advances in two existing benchmark TST tasks and one newly proposed challenging TST task. Human evaluations further confirm that LaMer generation is not only more readable but also richer in diverse expressions than previous methods. *Second*, We propose a new challenging text style transfer task, *Political Stance Transfer*, and release a benchmark dataset, which includes news sentence-pairs describing the same fact but from opposite political stances (i.e., *liberal* and *conservative*). The stances of the sentences are labeled by expert human editors. *Third*, As part of LaMer, we introduce a simple but powerful method to create parallel corpora from non-parallel data. We show that this method can be used as a general treatment to improve existing TST models. Given the abundance of non-parallel corpora for different TST tasks, this method has the potential to greatly impact the field of TST.

Code for LaMer is available at `https://github.com/DapangLiu/LaMer`.

## 2 APPROACH

LaMer is comprised of three main steps. First, we mine (roughly) parallels target style expressions for each of the source style sentences based on their LM-based sentence embeddings and scene graphs (§2.2). Second, we deploy MLE training on the text-to-text LM BART (Lewis et al., 2020b) to maximize the probability of generating such roughly parallel sentences in target style (§2.3). Finally, we use imitation learning to further refine the generation with the self-supervision from the mined parallel demonstrations (§2.4).

### 2.1 NOTATION AND PROBLEM STATEMENT

For the task of binary text style transfer (src ↔ tgt), We denote the source and target style datasets as $\mathcal{D}^{\text{src}}$ and $\mathcal{D}^{\text{tgt}}$, which contain $M$ and $N$ sentences expressing two different styles respectively (i.e., $\mathcal{D}^{\text{src}} = \{s_1^{\text{src}}, s_2^{\text{src}}, ...s_M^{\text{src}}\}$, $\mathcal{D}^{\text{tgt}} = \{s_1^{\text{tgt}}, s_2^{\text{tgt}}, ...s_N^{\text{tgt}}\}$). We also denote the tokens in source and target sentences as $\{x_1, x_2, ..., x_t\} \in s_i^{\text{src}}$ and $\{y_1, y_2, ..., y_t\} \in s_j^{\text{tgt}}$, for given source and target sentences $s_i^{\text{src}}$ and $s_j^{\text{tgt}}$. Given an input sentence $s^{\text{src}}$ of the source style, an ideal style transfer model should be able to generate an appropriate, fluent target sentence $\hat{s}^{\text{tgt}}$, which is of the target style while maintaining the style-independent content.

Existing methods learn a mapping $f_{\text{random}} : \mathcal{D}^{\text{src}} \to \mathcal{D}^{\text{tgt}}$ without considering the self-parallelism between $\mathcal{D}^{\text{src}}$ and $\mathcal{D}^{\text{tgt}}$; LaMer, however, works on the assumption that $m(\leq M)$ sentences in $\mathcal{D}^{\text{src}}$ have $n$ parallels in $\mathcal{D}^{\text{tgt}}$. The number of parallels $n$ for each of the $m$ sentences is variant but no more

than $N$. Therefore, the goal of LaMer can be framed as first identifying a parallel subset of the original non-parallel dataset and learning a mapping on this dataset: $f_{\text{parallel}} : \mathcal{D}_m^{\text{src}} \to \mathcal{D}_n^{\text{tgt}} (m \leq M, n \leq N)$.

## 2.2 MINING (ROUGHLY) PARALLEL SENTENCES

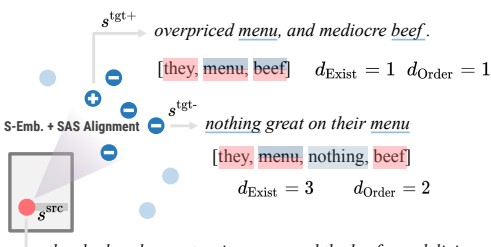

Figure 2: An example of aligned parallels by our scene graph based method. We _underline_ the scene entities and annotate how we compute $d_{\text{Order}}$ and $d_{\text{Exist}}$ with $\hat{s}^{\text{tgt+}}$ and $\hat{s}^{\text{tgt-}}$.

In this work, we argue that the *non-parallel* text style transfer datasets have inherent property of *parallelism*. We outline in the following three major strategies to mine roughly parallel sentences:

**Random (RD).** For each sentence $s^{\text{src}} \in \mathcal{D}^{\text{src}}$, randomly pick $k$ sentences from $\mathcal{D}^{\text{tgt}}$ as parallels. This method mainly serves as an ablation to show the effectiveness of other strategies.

**Sentence Embedding (S-Emb.).** Sentences that only differ in linguistic style should have higher semantic similarity than other unrelated sentences. As a more advanced strategy, we leverage off-the-shelf LMs (e.g., RoBERTa (Liu et al., 2019)) to mine semantically similar sentences. We first encode every sentence in $\mathcal{D}^{\text{src}}$ and $\mathcal{D}^{\text{tgt}}$ with their embeddings extracted from an LM. Then, for each sentence $s^{\text{src}} \in \mathcal{D}^{\text{src}}$, we pick $k$ most similar sentences from $\mathcal{D}^{\text{tgt}}$ through cosine similarity as parallels.

**S-Emb. + Scene Alignment Score (S-Emb. + SAS).** We add a scene graph based alignment step to the sentence embedding method to leverage shared entities between source and target sentences. Scene graph extraction has been successfully used in computer vision (Venugopalan et al., 2015; Chen et al., 2020b) to guide (Pan et al., 2020) or judge (Anderson et al., 2016) diverse caption generation. Inspired by the CV example and the observation that scene graphs of sentences with similar content but different styles tend to have overlapping nodes, our algorithm highlights the shared entities (nodes) of the sentence scene graph by running the scene graph parser released by Wu et al. (2019)[1], which outputs mined scenes graphs in **subject-relation-object** triplets. In our case, we only consider the end nodes of these triplets (which are *subj.* and *obj.*) as the *scenes entities*. Figure 2 shows an example of mined sentences and corresponding scenes entities.

Let $|\mathbf{e}(s^{\text{src}})|$ and $|\mathbf{e}(s^{\text{tgt}})|$ be the number of scene entities extracted from the source and target style sentence, and $|\mathbf{e}(s^{\text{src} \cap \text{tgt}})|$ be the number of their overlaps, we compute Scene Alignment Score (SAS) as:

$$\text{SAS}(s^{\text{src}} || s^{\text{tgt}}) = \frac{1}{|s^{\text{tgt}}|} \cdot \frac{(1 + \beta^2) \cdot \text{precision} \cdot \text{recall}}{\beta^2 \cdot \text{precision} + \text{recall}}, \tag{1}$$

where precision ($= \frac{|\mathbf{e}(s^{\text{src} \cap \text{tgt}})|}{|\mathbf{e}(s^{\text{tgt}})|}$) and recall ($= \frac{|\mathbf{e}(s^{\text{src} \cap \text{tgt}})|}{|\mathbf{e}(s^{\text{src}})|}$) describe the portion of target and source entities that are part of the overlap, and positive real factor $\beta$ is chosen such that recall is considered $\beta$ times as important as precision in a specific domain. We also normalize SAS by the length of $s^{\text{tgt}}$, since longer target sentence candidates are more likely to have higher SAS.

SAS serves as a complementary alignment metric to further filter the parallel sentences picked by the previous sentence-embedding-based metric. The complete procedure of S-Emb. + SAS alignment is:

1. Encode each sentence in $\mathcal{D}^{\text{src}}$ and $\mathcal{D}^{\text{tgt}}$ as sentence embeddings via a pre-trained LM.

2. Compute cosine similarity between all sentence pairs $(s_i, s_j)$ where $s_i \in \mathcal{D}^{\text{src}}$ and $s_j \in \mathcal{D}^{\text{tgt}}$. Pick the $k$ most similar sentences $s_j^{(k)}$ from $\mathcal{D}^{\text{tgt}}$ as parallel candidates for $s_i$.

---

[1]The scene graph parser can be found here: https://github.com/vacancy/SceneGraphParser

3. For each sentence $s_i \in \mathcal{D}^{\text{src}}$ and its parallel candidates $s_j^{(k)}$, compute pairwise SAS by Equation 1. Pick those $s_j$ whose SAS scores (with respect to $s_i$) are above a threshold $p$ as the final parallel sentences to $s_i$.

From empirical observations, we found that the selection of $p$ and $k$ is task-specific and has crucial effects on the performance of LaMer. We explore the outcomes of different choices of $k$ and $p$ in §3.3. Though SAS alignment LaMer relies on entity alignment for mining parallel corpora, other alignment mechanisms, such as (multilingual) LM alignment (i.e., the S-Emb. method) can potentially be used for transfer cases where entities may not be shared across styles (such as ancient v.s. modern English (Xu et al., 2012), or even multilingual text style transfer (Krishna et al., 2021)).

## 2.3 CONDITIONAL TOKEN-LEVEL MLE TRAINING

After mining parallel sentences for a source sentence $s^{\text{src}}$, from our corpus, we take the source sentence $s^{\text{src}}$ as the condition, and preform a conditional token-level MLE training on the mined parallels in the manner of standard autoregressive pre-training. Assuming for a certain source sentence $s^{\text{src}}$ we have $n$ target parallels mined in the previous step (i.e., $s^{\text{src}} \to \{s_1^{\text{tgt}}, s_2^{\text{tgt}}, ...s_n^{\text{tgt}}\}, n \leq N$), the MLE loss can be computed as:

$$J_{\text{MLE}} = -\sum_{i=1}^{n} \sum_{t=1}^{|s_k^{\text{tgt}}|} y_t^{(i)} \log \text{LM}(y_{1:t-1}|s^{\text{src}}), \qquad (2)$$

where $y_t^i \in s_j^{\text{tgt}}$ denotes the $i$-th step ground truth token belonging to the $i$-th target parallel sentence. $\text{LM}(y_{1:t-1}|s^{\text{src}})$ is the LM output at $t$-th step (a distribution on the vocabulary) given the source text and previous generation. We choose BART Lewis et al. (2020b) as the LM, which is already pre-trained on several text-to-text tasks, and concatenate the source and target text with `` tokens pre-defined by BART as input data. At each step, with the supervision of ground truth target tokens $y_t^{(i)}$, we minimize the MLE loss only at the target part. The LM is thus trained to maximize the probability of generating tokens $\hat{y}_t$ approaching to $y_t^{(i)}$.

Though the results produced are reasonable, solely relying on MLE training has several defects. First, MLE treats all parallels of a certain source equally without penalizing low-quality demonstrations, which not only makes the model less robust on noisy data, but also possible to be misled by negative samples during local optimization. Second, token-level MLE does not pay special attention to scene entities shared between the source and target, which we found take up the main portion in the human written references (as demonstrated in §A.3). We are thus motivated to further improve the performance using the self-supervision from the parallels.

## 2.4 REINFORCED IMITATION LEARNING BY CONTRAST

For all the mined target style parallels $s^{\text{tgt}}$ of $s^{\text{src}}$, we denote the one that has the highest similarity score (with S-Emb.) or has the highest SAS (with S-Emb. + SAS) as the expert demonstration $s^{\text{tgt+}}$, while the others are set as $s^{\text{tgt-}}$.

In the formulation of reinforcement learning, at each step $t$, the policy $\pi_\theta$ observes the state $y_{1:t-1}$ (partially generated text), takes an action $a_t = y_t$ (pick a token), and transits to the next state $y_{1:t}$. In the context of TST tasks, the policy can be interpreted as the LM-based generation model: $\hat{\pi} \sim \text{LM}(a_t|y_{1:t-1}, s^{\text{src}})$, and the set of $s^{\text{tgt+}}$ can be seen as the trajectory derived from an expert policy $\pi^+$, while $s^{\text{tgt-}}$ is produced by an amateur policy $\pi^-$. The goal of this step imitation learning can thus be framed as learning a policy that has more probability to behave like $\pi^+$ rather than $\pi^-$, which can be achieved by minimizing the contrastive loss:

$$J_{\text{IL}} = \max(\psi^* \left[\pi^+, \hat{\pi}\right] - \psi^* \left[\pi^-, \hat{\pi}\right] + \Delta, 0), \qquad (3)$$

where $\psi^*(\cdot)$ is the distance measurement of two policies in terms of semantic coherence $d_{\text{SEM}}(\cdot)$ and scene preservation $d_{\text{PSV}}(\cdot)$, such that $\psi^*(\cdot) = d_{\text{SEM}}(\cdot) + d_{\text{PSV}}(\cdot)$; and $\Delta$ is a hyperparameter

that controls the contrasting margin. The general form is inspired by the triplet loss (Schroff et al., 2015) and recent *in-batch* constrastive learning (Gao et al., 2021): we encourage the learned policy $\hat{\pi}$ to be more similar to the expert one $\pi^+$ than the amateur one $\pi^-$ by at least margin $\Delta$. $d_{\text{SEM}}(\cdot)$ and $d_{\text{PSV}}(\cdot)$ are detailed below.

**Sequence-level Semantic Coherence.** For sequence $\hat{s}$ generated under current learning policy $\hat{\pi}$ and a certain target style demonstration $s^{\text{tgt}}$, we encode them using bi-directional LM RoBERTa (Liu et al., 2019), which is widely used as sentence encoder in related work (Reimers & Gurevych, 2019; Li et al., 2020). Then, we take their negative cosine similarity as the distance for semantic coherence: $d_{\text{SEM}} = -\frac{\text{emb}_{\hat{s}}^{\text{T}} \cdot \text{emb}_{s^{\text{tgt}}}}{\|\text{emb}_{\hat{s}}\| \|\text{emb}_{s^{\text{tgt}}}\|}$. Lower value in $d_{\text{SEM}}$ means the learned policy $\hat{\pi}$ can produce similar quality sequences in terms of semantics as target style demonstrations. It is a sequence-level measurement since the encoding is based on the whole sentence.

**Token-level Scene Preservation.** In our context, legitimate target style parallels should preserve most scene entities that appear in source style sentences. To explicitly promote the content preservation performance of the current policy, we introduce a token-level distance $d_{\text{PSV}}$. We first extract scene entities from $s^{\text{tgt+}}$ and each of the $s^{\text{tgt-}}$, and then measure the differences in both the order of appearance of the entities and the existence of the entities. Taking $s^{\text{tgt+}}$ as an example (similar calculation is done on each of the $s^{\text{tgt-}}$), we compute: $d_{\text{PSV+}} = \alpha d_{\text{Order+}} + (1-\alpha)d_{\text{Exist+}} = \alpha\|\vec{e}(s^{\text{tgt+}}), \vec{e}(s^{\text{src}})\|_{\text{Lev.}} + (1-\alpha)\|\vec{e}(s^{\text{tgt+}}) - \vec{e}(s^{\text{tgt+} \cup \text{src}})\|_H^2$ where $\vec{e}(\cdot)$ means the extracted scene entities, $\vec{e}(\cdot)$ is its one-hot encoded vector, $\|\vec{x}, \vec{y}\|_{\text{Lev.}}$ measures Levenshtein distance (Yujian & Bo, 2007), one type of edit distance, and for vectors $\vec{e}(s^{\text{src}})$ and $\vec{e}(s^{\text{tgt}})$, we compute their hamming distance[3] (Norouzi et al., 2012). Lower $d_{\text{Order+}}$ means the appearing order of $\vec{e}(s^{\text{src}})$ resemble $\vec{e}(s^{\text{tgt+}})$. Similarly, lower $d_{\text{Exist+}}$ means $\vec{e}(s^{\text{src}})$ and $\vec{e}(s^{\text{tgt+}})$ are mostly overlapped. Similar procedure applies to $d_{\text{PSV-}}$ as well. We present more details about how we compute these distances in §A.4.

### 2.4.1 POLICY GRADIENT BY REINFORCE

We choose the REINFORCE algorithm (Williams, 1992) to optimize the current policy $\pi_\theta$. As shown in Equation 4, we use greedy decoding results as the baseline (Rennie et al., 2017), and sampling decoding for the exploration step, which mimics the *actor-critic* manner in reinforcement learning (Fujimoto et al., 2018; Haarnoja et al., 2018). The policy gradient can thus be presented as:

$$\nabla_\theta J(\theta) = -\mathbb{E}_{\pi_\theta}\left[ \nabla_\theta \log \pi_\theta \cdot (J_{\text{IL}}^{\text{sample}}(s^{\text{tgt+}}, s^{\text{tgt-}}) - J_{\text{IL}}^{\text{greedy}}(s^{\text{tgt+}}, s^{\text{tgt-}})) \right], \tag{4}$$

where $J_{\text{IL}}^{\text{sample}}$ and $J_{\text{IL}}^{\text{greedy}}$ can be both calculated by Equation 3. We take the difference of two the mode rewards as the advantage, feeding into the REINFORCE algorithm to optimize our policy $\pi_\theta$. Finally, the output policy $\pi_\theta$ is able to generate style transferred sequences resembling the expert demonstrations $s^{\text{tgt+}}$. More details about the complete loop of policy gradient can be found in §A.5

**Discussion: Imitation Learning, Reward-based RL, and MLE.** Framing token-by-token text generation as a sequential decision-making process, reward-based reinforcement learning (RL) has achieved success in text generation by directly setting indifferentiable evaluation metrics such as BLEU (Papineni et al., 2002) or ROUGE (Lin, 2004) as the reward function (Gong et al., 2019; Liu et al., 2020b). In the context of TST tasks, however, instead of reward-based RL, we argue imitation learning (IL) is a better choice, since: 1) it is challenging to design a uniform reward fitting a variety of source style text (Chen et al., 2020a), and 2) it is also unnecessary to calculate an elaborately designed reward. The roughly parallel target style sentences mined in the alignment step can naturally serve as expert demonstrations (Li et al., 2018b). They are high-quality and cost-free, which can greatly improve the sampling and learning efficiency of RL optimization.

A common obstacle of IL is known as *covariate shift* (Ross & Bagnell, 2010), which means IL can provide a relatively good performance for the test cases similar to the demonstrations (used for policy training), but it could still suffer from bad generalization for cases it never meets during training, as

---

[2] $\alpha$ controls the weights assigned to $d_{\text{Order}}$ and $d_{\text{Exist}}$; set by running repeated experiments ranging the $\alpha$ from 0 to 1 by 0.1, and picking the best-performing $\alpha$ with respect to GM: $\alpha = \{0.4, 0.3, 0.1\}$ for the three tasks.

[3] Hamming norm $\|\vec{v}\|_H$ is defined as the number of non-zero entries of vector $\vec{v}$.

the number of demonstrations is finite. Similar problems called *exposure bias* (Bengio et al., 2015; Ranzato et al., 2016) also exists in MLE when the supervised data is scarce. We claim our token-level MLE training on LMs can somewhat mitigate such problems, since the large-scale LMs are already pre-trained on huge quantities of data across many domains *as a prior*.

## 3 EXPERIMENTS

### 3.1 DATASETS AND EXPERIMENTAL SETUP

**Sentiment Transfer.** We use the Yelp reviews dataset collected by Shen et al. (2017) which contains 250k negative sentences and 380k positive sentences, organized in non-parallel fashion. Li et al. (2018a) released a small test set that has 500 human written references for both target sentiments, and they are already one-on-one aligned with the source sentiment sentences.

**Formality Transfer.** A more challenging TST task is to modify the formality of a given sentence. We use the GYAFC dataset (Rao & Tetreault, 2018), which contains formal and informal sentences from two domains. In this paper, we choose the Family Relationship domain, which consists of about 52k training sentences, 5k development sentences, and 2.5k test sentences. Different from the Yelp dataset, the test set of GYAFC has four human written references for each source style test case. Our evaluation results for this task are reported as the average against the four references.

**Political Stance Transfer.** News outlets reporting the same event naturally meet the requirement of style transfer data, which is aligned in content but different in style. We leverage Allsides.com[4] as an aggregator, where outlets reporting the same events with different political stances are already organized and labeled by expert human editors. In total, we collect 2,298 full-length news articles from 6/1/2012 to 4/1/2021 in pairs of liberal-conservative leaning. Though the data seems parallel at the article level, in this work we focus on sentence level TST. We further split the articles into about 56k sentence pairs as a non-parallel dataset for training (detailed in §A.2). These titles are written by expert writers from liberal- or conservative-learning media outlets, and aligned by the Allsides editors. For ethical consideration, we manually examined the articles to removing those containing sensitive topics (such as hate speech). To adhere to the outlets' TOS, we will release our dataset as links to the articles (instead of the text), with the appropriate code to extract the content for training purposes. A closer comparison of our proposed novel dataset with another political stance TST dataset (Voigt et al., 2018) can be found in §A.1.

The detailed statistics of all three benchmark datasets and our experimental setup can be found in §A.3. We have discussed the hyperparameter selection for the best performing LaMer in §3.3.

### 3.2 MAIN RESULTS ON TEXT STYLE TRANSFER PERFORMANCE

TST models should be able to transfer source style sentence to a target style, while maintaining the style-independent content as much as possible. Most existing works evaluate their methods with transfer accuracy (ACC; classification accuracy) and content-preservation (BLEU-*ref*; token overlaps with reference). In this work, besides using ACC and BLEU-*ref*, we also report more advanced metrics–SIM and FL introduced by Krishna et al. (2020)–to further evaluate our method.

**LaMer has competitive performance.** Table 1 shows the main results comparing LaMer with eight popular TST models on the three benchmark datasets. In addition to the independent ACC and BLEU score[5], we also report their geometric mean (GM) for direct comparison with the DualRL and STRAP baselines; however, it should be noted that GM has not been well studied as an evaluation metric for TST and is only included for comparison purposes. We also consider inverse PINC scores (Chen & Dolan, 2011)—$i$-PINC—as a complementary metric of BLEU. $i$-PINC counts the n-gram overlaps after removing the tokens shared between source and target style sentences, and thus properly penalizes the Input Copy method ($i$-PINC = 0).

Compared with existing methods, LaMer performs better in terms of overall metric GM, and shows particularly strong *net* transfer ability (since $i$-PINC does not count duplicated tokens between $s^{\text{src}}$

---

[4] https://www.allsides.com/story/admin
[5] We use BLEU-*ref*, which computes average n-gram BLEU (n ranges from 1 to 4).

Table 1: LaMer achieves competitive performance in three TST tasks. CAE (Shen et al., 2017), DelRetrGen (Li et al., 2018a), DualRL (Luo et al., 2019), Style Transformer (Dai et al., 2019), Deep Latent Seq (He et al., 2020) are popular TST models *not* built on LMs. TSF-DelRetGen (Sudhakar et al., 2019), IMaT (Jin et al., 2019), and STRAP (Krishna et al., 2020) are recent LM-based TST models. Input Copy directly copies the source style sentence as the transfer output. GM: Geometric Mean of Transfer Accuracy (ACC) and BLEU. *i*-PINC measures net transfer ability by not counting words that can be directly copied from the source style sentence, as discussed in §3.2. We color (■ ■ ■) the best results, and underline the second best, not including the Input Copy reference.

| | ■ **Sentiment** SAS = 0.73 | | | | ■ **Formality** SAS = 0.29 | | | | ■ **Political Stance** SAS = 0.30 | | | |
|---|---|---|---|---|---|---|---|---|---|---|---|---|
| **Existing Methods** | ACC | BLEU | GM | *i*-PINC | ACC | BLEU | GM | *i*-PINC | ACC | BLEU | GM | *i*-PINC |
| Input Copy (*ref.*) | 7.4 | 61.7 | 21.4 | 0 | 1.9 | 54.6 | 10.2 | 0 | 7.5 | 17.8 | 11.6 | 0 |
| CAE | 70.5 | 6.0 | 20.6 | 1.2 | 39.5 | 35.8 | 37.6 | 5.9 | 31.5 | 5.1 | 12.7 | 1.4 |
| DelRetrGen | 65.4 | 11.8 | 27.8 | 4.8 | 61.9 | 14.8 | 30.3 | 6.6 | 93.8 | 3.5 | 18.1 | 6.2 |
| Dual RL | 85.6 | 21.2 | 42.5 | 4.0 | 71.1 | 31.9 | 47.9 | 7.2 | 91.8 | 8.1 | 27.3 | 5.7 |
| Style Transformer | 93.6 | 17.1 | 40.0 | 5.2 | 72.2 | 32.7 | 48.6 | 4.1 | 48.6 | 7.3 | 18.8 | 1.1 |
| Deep Latent Seq | 87.9 | 18.7 | 40.5 | 7.3 | 80.5 | 18.5 | 38.6 | 6.1 | 85.8 | 20.8 | 42.2 | 3.0 |
| **LM-based** | | | | | | | | | | | | |
| TSF-DelRetGen | 89.5 | 31.7 | 53.3 | 4.5 | 79.3 | 17.4 | 37.1 | 7.3 | 65.8 | 23.7 | 39.5 | 2.9 |
| IMaT | 95.9 | 22.5 | 46.5 | 3.3 | 72.1 | 28.2 | 45.1 | 12.7 | 82.9 | 19.5 | 40.2 | 5.3 |
| STRAP | 92.3 | 32.5 | 54.8 | 6.4 | 75.4 | 32.6 | 49.6 | 9.7 | 84.3 | 21.3 | 42.4 | 5.0 |
| **Ours**: LaMer | | | | | | | | | | | | |
| w/. RD | 97.1 | 19.8 | 43.8 | 2.5 | 64.2 | 20.4 | 36.2 | 5.5 | 75.9 | 10.1 | 27.7 | 6.1 |
| w/. S-Emb. | 90.1 | 40.6 | 60.5 | 5.6 | 65.1 | 33.8 | 46.9 | 10.9 | 78.2 | 22.1 | 41.6 | 9.2 |
| w/. S-Emb. + SAS | 97.0 | 34.1 | 57.5 | 9.6 | 76.5 | 39.2 | 54.8 | 13.3 | 82.7 | 30.5 | 50.2 | 13.6 |

and $s^{\text{tgt}}$). Among the three mining strategies we proposed in §2.2, the S-Emb. + SAS seems to offer the best performance boosting in terms of content preservation, but shows no clear advantage on style control accuracy. When comparing LaMer with the three LM-based models, we find the advantage of LaMer mainly comes from the gain in content preservation (BLEU), which demonstrates the effectiveness of our scene graph alignment. We also find some of the other TST models produce low-fluency (i.e., high perplexity) transferred sentences (see Table 10 in §A.7), though we must note that recent studies have shown perplexity to be a poor measure of fluency (Krishna et al., 2020; Mir et al., 2019; Lee et al., 2021).

Table 2: Few-shot performance on Formality TST (only 1% training data available) of LaMer and other LM-based baselines. We also include a zero-shot TST method built on GPT-3 (Reif et al., 2021) for reference. We annotate standard deviation next to the results as well.

| Training Data (%) | 1% (0% for GPT3) | | | | 100% | | | |
|---|---|---|---|---|---|---|---|---|
| **Method** | ACC | SIM | FL | $J_{\textbf{A,S,F}}$ | ACC | SIM | FL | $J_{\textbf{A,S,F}}$ |
| TSF-DelRetGen | $16.5_{1.4}$ | $5.5_{1.9}$ | $90.3_{0.1}$ | $2.6_{1.1}$ | $54.1_{1.4}$ | $30.5_{0.8}$ | $98.9_{0.1}$ | $29.7_{0.5}$ |
| IMaT | $35.9_{0.4}$ | $22.1_{0.2}$ | $23.0_{0.4}$ | $3.2_{0.7}$ | $59.5_{1.2}$ | $39.7_{1.1}$ | $99.6_{0.1}$ | $37.4_{0.2}$ |
| STRAP | $37.7_{0.7}$ | $12.5_{0.9}$ | $70.4_{0.2}$ | $5.3_{0.4}$ | $67.7_{1.2}$ | $72.5_{1.1}$ | $90.4_{0.7}$ | $45.5_{1.2}$ |
| **Ours**: LaMer | $48.8_{0.4}$ | $27.4_{0.6}$ | $89.8_{0.1}$ | $\mathbf{13.5_{0.3}}$ | $66.4_{0.5}$ | $74.3_{0.3}$ | $97.8_{0.1}$ | $\mathbf{53.2_{0.1}}$ |
| GPT-3$_{\text{prompt TST}}$ | $10.2_{0.6}$ | $17.1_{1.1}$ | $98.0_{0.2}$ | $6.5_{0.4}$ | – | – | – | – |

Table 3: Compared with other baselines, LaMer requires neither extra data nor training additional models.

| Ext. Resource? | Data | Model |
|---|---|---|
| TSF-DelRetGen | ✓ | ✓ |
| IMaT | ✓ | ✗ |
| STRAP | ✓ | ✓ |
| **Ours**: LaMer | ✗ | ✗ |
| GPT-3$_{\text{prompt TST}}$ | ✗ | ✗ |

**LaMer is robust in few-shot settings.** LaMer learns the inherent parallelism within the unsupervised data, which could be challenging in data scarce cases. Recent works have explored the limits of TST in few-shot settings (Riley et al., 2021; Krishna et al., 2021), we conduct similar studies by limiting the available training data (to 1%), aiming to answer whether or not LaMer can still provide comparable performance as others. We consider LM-based baselines which show strong results in Table 1. We also compare with a zero-shot prompt-based TST baseline built upon GPT-3 (Reif et al., 2021) for reference. For evaluation metrics, we use the recently proposed *J*-score framework (including ACCuracy, SIMilarity, and FLuency) (Krishna et al., 2020), which solves many of the

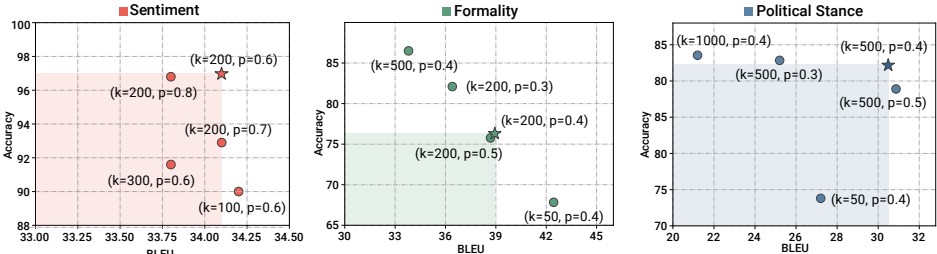

Figure 3: Effect on performance when choosing different $k$ and $p$ for S-Emb. + SAS alignment. We also annotate (with a star) the recommended settings which reaches best balance between style control accuracy and content preservation BLEU score.

issues in existing metrics [6]. As shown in Table 2, we find that LaMer achieves great performance ($J$-score) in few-shot cases, partially because other baselines rely on external systems trained on ample data;[7] In extreme data hungry cases, the compromised external modules would propagate errors to following parts as they are coupled. LaMer, instead, requires neither extra data nor additional training on external systems (See Table 3), but fully leverages the self-supervision hidden in the unsupervised data, which achieves robust performance even in few-shot scenarios.

**IL refinement is crucial for LaMer content preservation.** As shown in Table 4, the BLEU score for LaMer drops substantially when the imitation learning refinement stage is totally removed (No IL, MLE only), and we find this drop is higher for challenging style tasks (such as political stance transfer), partially because the quality variance of their demonstrations is higher, i.e., a further guidance towards good demonstrations will make a big difference. We also conducted further ablation studies on components of IL: In general, removing $d_{\mathrm{Order}}$ seems to cause greater performance drop in formality TST, while $d_{\mathrm{Exist}}$ seems important to successful political stance TST. Removing all scene preservation distance sub-components (i.e., IL ($d_{\mathrm{SEM}}$)) leads to noticeable performance deterioration across all three TST tasks. We take these ablation study results as evidence that the technologies we propose for LaMer are all crucial.

Table 4: Further ablation on components of LaMer imitation learning (IL). $d_{\mathrm{SEM}}$: semantic coherence distance, $d_{\mathrm{Order}}$ and $d_{\mathrm{Exist}}$ are sub-components of scene preservation distance $d_{\mathrm{PSV}}$. Removing all IL modules (MLE only) will cause severe drop in content preservation (BLEU), especially for challenging TST tasks (Formality and Political Stance TST).

| | ■ **Sentiment** | | ■ **Formality** | | ■ **Political Stance** | |
|---|---|---|---|---|---|---|
| Ablation on LaMer IL | ACC | BLEU | ACC | BLEU | ACC | BLEU |
| IL ($d_{\mathrm{SEM}} + d_{\mathrm{Order}} + d_{\mathrm{Exist}}$) | 97.0 | 34.1 | 76.5 | 39.2 | 93.4 | 18.5 |
| IL ($d_{\mathrm{SEM}} + d_{\mathrm{Order}}$) | 95.5 (↓1.5) | 34.4 (↑0.3) | 74.7 (↓1.8) | 37.7 (↓1.5) | 93.2 (↓0.2) | 15.8 (↓2.7) |
| IL ($d_{\mathrm{SEM}} + d_{\mathrm{Exist}}$) | 96.5 (↓0.5) | 33.9 (↓0.2) | 72.6 (↓3.9) | 34.1 (↓5.1) | 92.7 (↓0.7) | 17.1 (↓1.4) |
| IL ($d_{\mathrm{SEM}}$) | 95.1 (↓1.9) | 33.3 (↓0.8) | 71.9 (↓4.6) | 32.5 (↓6.7) | 89.9 (↓0.5) | 13.9 (↓4.6) |
| No IL (MLE Only) | 93.1 (↓3.9) | 33.2 (↓0.9) | 68.7 (↓7.8) | 27.2 (↓12.0) | 89.8 (↓0.6) | 8.8 (↓9.7) |

## 3.3 Configuring LaMer

The filtering parameter $p$ and $k$ are hyperparameters that are crucial for the construction of roughly parallel datasets. Thus, for all three TST tasks, we evaluate LaMer performance given different combination of $p$ and $k$, as shown in Figure 3. We find there is a approximate trade-off relationship between style control accuracy and content preservation BLEU score: In general, higher $k$ (more parallels) seems to improve accuracy but normally has lower BLEU, while for a fixed $k$, a proper

---

[6]E.g., BLEU score can be artificially high when directly copying source style sentence as the target style generation, and perplexity is unbounded and discourages creative TST generation.

[7]E.g., TSF-DelRetGen requires training a classifier to locate style-carrying spans of text. STRAP fine-tunes a GPT-2 model to learn how to paraphrase. IMaT trains a plain Seq2Seq model from scratch as an LM.

$p$ can be reached empirically. We annotate the recommended settings of $p$ and $k$ for the three TST tasks we studied in this work.

## 3.4 HUMAN EVALUATION

We conducted human evaluation to answer: 1) *"How good is LaMer's SAS alignment?"*, and 2) *"How is the quality of the style-transferred sentences?"* (The exact questions and procedures are details in §A.6.). We recruited 212 participants from MTurk. Most annotators agreed that SAS-aligned pairs share similar style-independent content (Table 5). We conducted several one-way ANOVAs to test the difference among the seven baselines and LaMer (political stance TST results shown in Table 6; see Table 8 for sentiment TST and Table 9 for formality TST in §A.6). LaMer was significantly better than next-best in all three perspectives ($p < .05$), and had the most ratings that where $\geq 5$.

Table 5: Effectiveness of SAS alignment for mining weakly parallel sentences. Similarity: *"How much the mined pairs are similar in style-independent content?"* SAS Align: *"How well SAS scores reflect such similarity?"* Ratings are on a 7 point scale. All$_{\geq 4}$: The ratio for which the ratings are $\geq 4$ for both questions.

| | Similarity | SAS | |
|---|---|---|---|
| TST Tasks | Mean$_{SD}$ | Mean$_{SD}$ | All$_{\geq 4}$ |
| Sentiment | $5.42_{1.36}$ | $5.56_{1.17}$ | 88% |
| Formality | $5.58_{1.14}$ | $5.74_{1.09}$ | 82% |
| Political | $5.45_{1.17}$ | $5.43_{1.16}$ | 74% |

Table 6: Human evaluations on the political stance transfer task (on a 7 point scale). We **bold** the highest average rating. LaMer was rated significantly higher than the next-best method ($p < .05$). All$_{\geq 5}$: The ratio for which the ratings are $\geq 5$ for all three perspectives.

| Political Stance | Style | | Content | | Readability | | |
|---|---|---|---|---|---|---|---|
| Methods | Mean | SD | Mean | SD | Mean | SD | All$_{\geq 5}$ |
| CAE | 4.91 | 1.59 | 4.85 | 1.58 | 5.04 | 1.68 | 12% |
| DelRetrGen | 5.14 | 1.60 | 4.93 | 1.63 | 4.96 | 1.60 | 11% |
| Dual RL | 5.15 | 1.37 | 4.87 | 1.59 | 5.15 | 1.53 | 27% |
| Style Transformer | 5.31 | 1.19 | 5.34 | 1.36 | 5.18 | 1.40 | 28% |
| Deep Latent Seq | 5.42 | 0.57 | 5.29 | 1.20 | 5.30 | 1.26 | 25% |
| TSF-DelRetGen | 4.87 | 0.65 | 5.23 | 1.14 | 5.20 | 1.13 | 15% |
| STARP | 5.33 | 1.25 | 5.31 | 1.40 | 5.43 | 1.07 | 21% |
| **Ours**: LaMer | **5.59** | 1.17 | **5.44** | 1.11 | **5.56** | 1.05 | 37% |

## 4 RELATED WORK

Besides the baselines we have studies in the paper, we notice there are other TST methods focusing on either latent vector manipulation (Liu et al., 2020a; Wang et al., 2019; John et al., 2019) or surface level editing (Castro et al., 2017; Karadzhov et al., 2017; Mansoorizadeh et al.). LaMer differs from them as it is learning to transfer with self-supervision mined from the unsupervised data, without complicated manipulation or editing. The idea of mining roughly parallel demonstrations in LaMer seems to echo the recent findings of retrieval-based generation systems (Lewis et al., 2020a; Lee et al., 2019) — retrieving relevant documents from external resources (e.g., Wikipedia) as grounding can improve the quality of generation (Lewis et al., 2020c; Guu et al., 2020). LaMer does not rely on any external resources, but mines parallel demonstrations from the training data itself, and thus is easier to deploy and alleviates potential bias.

## 5 CONCLUSION

We have proposed LaMer, a self-supervised text style transfer method built on large-scale LMs. LaMer demonstrates the potential of leveraging self-parallelism within large non-parallel corpora, which can guide efficient TST models development or even LM design. On three diverse TST tasks, we show that LaMer performs better than eight existing methods while being robust to data scarcity. Human evaluations further validate the advantage of our model compared to existing baselines.

Future work of LaMer could study how to extend to multilingual scenarios, or transfer the style that essentially changes the form of language (e.g., ancient v.s. modern English). One could also investigate combining the alignment module of LaMer with the text2text LM in one model, and jointly train to further boost the performance. Another direction is to further study the integration of LaMer with larger size foundation models, like GPT-3 175B to understand LaMers potential in few-shot settings.

## 6 ETHICS AND REPRODUCIBILITY STATEMENT

The goal of LaMer is to provide a simple but efficient general-purpose text style transfer framework by leveraging large-scale pre-trained LMs. LaMer can generate text in a specified target style given source style input, which can serve as a useful tool for social good. For example, LaMer can assist language formalization (formality transfer), enable data anonymization (writing style transfer), or depolarize politically biased text (Liu et al., 2022; 2021a), via transferring the identifiable style of the input text.

The performance of LaMer depends on the pre-trained LM, which means that the generation of LaMer can be affected by certain societal biases found in the pre-trained LM it is based on (e.g., BART) (Blodgett et al., 2020). Other negatives such as abusive language (Hancock et al., 2020), hate speech (Kennedy et al., 2018), and dehumanization (Mendelsohn et al., 2020) are also prevalent in the language data for LM pre-training. When deployed in real-world settings, LaMer may increase the risk of reinforcing and propagating polarizing and toxic language in public space through people's dissemination (Waseem et al., 2017). Moreover, the generated text of LaMer could potentially be included in the next iteration of large-scale LM training data collection, which may amplify offensive stereotypes and overtly abusive views in the future LMs.

Though LaMer was developed as an "neutral" tool mainly for research purposes, it can also be used in malicious ways such as generating sophisticated spam in a particular style "disguise", which we strongly discourage (Rae et al., 2021; Liu et al., 2021b). Due to the lack of fact checking systems as a safety measure, LaMer is at the risk of generating seemingly coherent but counterfactual text in a certain target style, which could further undermine social trust in computer-mediated forms of communication (Liu et al., 2021c; Petroni et al., 2019). We thus recommend the users of LaMer to integrate necessary safety checks into the actual systems.

In addition, our experiments and analysis are done when source and target style sentences are both in English, and therefore we do not claim that our findings will generalize across all languages, although our framework has the potential to be extended to other languages with necessary modifications.

Though it's beyond the scope of this paper and our individual ability to address all above-mentioned complications, we strongly encourage users of our system (and other TST models and systems based on pre-trained language models) to acknowledge and beware of these issues.

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

# A    APPENDIX

## A.1    COMPARING WITH ANOTHER POLITICAL STANCE DATASET

Transferring political stance has also been studied in Voigt et al. (2018), where the author also collected a dataset for political stance TST. Comparing with their dataset, we found ours has following differences:

**Language Formality**: Their dataset is composed of comments on Facebook posts from members of the United States Senate and House, while our dataset consists of sentences from news articles written by professional reporters belonging to well-known media outlets (CNN, Fox News, etc.). The text in the dataset of Voigt et al. (2018) tends to be more casual and oral (e.g., *"congrats , lee !"*). Conversely, the text in our dataset is generally more formal and well-formed.

**Completeness**: Their dataset does not contain the original post to which the comments are responding, which causes some potential ambiguity on its political stance labels especially when the comment is generic (e.g., *"happy birthday jerry : - )"*). Most of the sentences in our dataset are complete statements of certain facts. We also conducted an extra distillation step (described in Appendix A in the supplementary document) to only keep the sentences that are salient to the article's stance label.

**Coverage**: The comments in Voigt et al. (2018) are mainly from 2017-2018, while our dataset collects nearly 10-year of articles from 6/1/2012 to 4/1/2021, which covers a wider range of societal issues and events. In the released version of our data we will add the time stamps to the sentences for customized filtering.

## A.2    SENTENCE DISTILLATION FOR CONSTRUCTING POLITICAL STANCE DATASET

As mentioned in Section 3.1 of the paper, the articles collected from Allsides.com are already manually aligned (based on their political ideology) by Allsides editors. We collected a total of 2,298 full-length news articles from opposite ideologies (i.e., *liberal* and *conservative*) grouped by events. To create a sentence-level TST dataset for political stance transfer, we split the articles into sentences and took their article-level ideologies (as assigned by the editors) as their sentence-level stance labels. However, we found that not all sentences explicitly exhibit the stance label inherited from their corresponding articles, so we implemented an additional distillation step to select those sentences that are salient with respect to their assigned political stance.

Assuming we have an article with $N$ sentences $X = \{x_1, x_2, ..., x_N\}$ and the title of the article $x_{\text{title}}$, we aim to find the subset of sentences $D_{\text{distill}} \subset X$ which all explicitly exhibit the assigned political stance. To better describe our approach, we first introduce two concepts: (1) stance saliency, and (2) distillation cost.

**Stance Saliency.**    We compute the stance saliency $v_i$ for each sentence $x_i \in X \backslash D_{\text{distill}}$, which is a measure of how well $x_i$ exhibits a certain political stance. $D_{\text{distill}}$ is the set of sentences already picked our distilled set. We assume that the stance exhibited in $x_{\text{title}}$ matches that of the article (as assigned by the editors), Thus, we define stance saliency $v_i$ as the similarity between $x_i$ and the union of $x_{\text{title}}$ and $D_{\text{distill}}$, which is:

$$v_i := \text{LCS}(x_i, x_{\text{title}} \cup D_{\text{distill}}), \tag{5}$$

where $\text{LCS}(\cdot)$ is longest common sequence between the two input sequences. However, this metric is biased towards longer sentences as they can have more tokens matched with the title. We address this issue through the intriduction of a distilation cost, described below.

**Distillation Cost.**    We assign a distillation cost $w_i$ to each sentence $x_i$ in an article. The cost is simply the length of the sentence:

$$w_i := |x_i|. \tag{6}$$

Note that for both concepts our computation is based on lower-cased and lemmatized sentence tokens. We use $\lambda$ to denote the ratio of the sentences to be picked as distilled sentences in an article consisting of $N$ sentences, and define $W_{\max} = \lambda \cdot \sum_{i=0}^{N} w_i$ as the maximum distillation cost allowed.

Now that for each sentence we have a stance saliency and a distillation cost, we aim to choose a set of sentences as the distillation which have the highest possible sum of saliency for which the sum of distillation costs is not greater than $W_{\max}$. Given a function $\phi(x_i) = \{1, 0\}$ where 1 corresponds to sentence $x_i$ being picked for the distilled set and 0 means otherwise, the objective of distillation can be expressed as:

$$\max \sum_{i=1}^{N} v_i \cdot \phi(x_i) , \quad \text{s.t.} \sum_{i=1}^{N} w_i \cdot \phi(x_i) \leq W_{\max} . \tag{7}$$

This is actually a NP-complete combinatorial optimization problem, called the Knapsack problem, which we solve using dynamic-programming (DP). In general, the distillation strategy aggressively picks sentences of high stance saliency while keeping the cost of distillation from exceeding the mask cost limitation $W_{\max}$. The procedure is demonstrated in Algorithm 1 below.

---

**Algorithm 1:** Sentence Distillation for Political Stance Dataset

**Input:** article title $x_{\text{title}}$, article sentences $X$, distillation ratio $\lambda$.

▷ Initialization
Init current distillation set $D_{\text{distill}}$ as $\varnothing$;
Init current distillation cost $W_{\text{distill}}$ with $|x_{\text{title}}|$;

▷ Multiple Rounds of Distillation
**while** $W_{distill} \leq W_{max}$ **do**
    Compute $v_i$ for each $x_i \in X \backslash D_{\text{distill}}$ with $x_{\text{title}}$ and $D_{\text{distill}}$ (by Equation 1);
    Compute $w_i$ for each $x_i$ (by Equation 2);
    Find best distillation strategy $\{\phi(x_i)\}$ via DP with $v_i$, $w_i$ and $W_{\text{distill}}$ (see Alg. 2);

    ▷ Update Distillation Results Every Step
    Update current distillation set $D_{\text{distill}} = D_{\text{distill}} \cup \{x_i\}$ whose $\phi(x_i) = 1$;
    Update current distillation cost $W_{\text{distill}}$ in terms of updated $D_{\text{distill}}$;
**end**
**return** distillation sentences set $D_{\text{distill}}$;

---

Note that we initialize $W_{\text{distill}}$ with the length of title. At every iteration of the loop we solve the DP problem with the limit set by current $W_{\text{distill}}$, which is a variable that estimates how many sentences are already in the distillation set (no more than $W_{\max}$). The DP-based algorithm used in Algorithm 1 for finding the best distillation strategy is shown is Algorithm 2 below.

### A.3 Statistics our All Three Datasets, and Experimental Setup

We present more statistical details of all three datasets in Table 7. All of our experiments were run on a single RTX-2080 GPU, with batch size 4 and 2/3/2 epochs for LaMer in the above three TST tasks. Following previous work, we choose style control accuracy (ACC), BLEU score as our main metrics. ACCs are judged by a pre-trained RoBERTa (Liu et al., 2019) classifier with F1 = {0.97, 0.86, 0.93} using the training data of the three TST tasks. About the $J$-score we calculate in few-shot experiments, we use the models released by the author[8].

### A.4 Additional Details about Token-level Scene Preservation

In this section, we provide additional details regarding how we compute the token-level scene preservation $d_{\text{PSV}}$. Figure 4 shows an example sentiment transfer task. For a given positive (source style) sentence $s^{\text{src}}$ and its corresponding negative (target style) parallels $s^{\text{tgt}}$, we first extract their

---

[8] The evaluation scripts, and pre-trained models can be found here: http://style.cs.umass.edu/

| Training Data (S-Emb. + SAS) | ■ **Sentiment** | | ■ **Formality** | | ■ **Political Stance** | |
|---|---|---|---|---|---|---|
| | *positive* | *negative* | *formal* | *informal* | *liberal* | *conservative* |
| # Sent Pairs | 553,430 | | 93,157 | | 36,307 | |
| # Scene Entities | 742,136 | 777,839 | 145,731 | 133,779 | 194,032 | 272,071 |
| Avg. Sent Len | 7.32 | 7.58 | 11.52 | 11.05 | 19.39 | 24.75 |
| Avg. # Scene Ent. / Sent | 1.34 | 1.41 | 1.56 | 1.44 | 5.34 | 7.49 |
| Avg. # Scene Ent.$^{\text{src}\cap\text{tgt}}$ (vs. RD) | 0.95 (0.73) | | 0.49 (0.02) | | 0.96 (0.10) | |
| Avg. SAS (vs. RD) | 0.72 (0.55) | | 0.29 (0.01) | | 0.25 (0.01) | |
| **Human Reference** | *positive* | *negative* | *formal* | *informal* | *liberal* | *conservative* |
| # Sent Pairs | 500 | | 1020 | | 525 | |
| # Scene Entities | 964 | 905 | 1,467 | 1,388 | 2,915 | 3,012 |
| Avg. Sent Len | 9.90 | 9.80 | 10.9 | 9.35 | 10.30 | 11.19 |
| Avg. # Scene Ent. / Sent | 1.93 | 1.81 | 1.44 | 1.36 | 5.58 | 5.76 |
| Avg. # Scene Ent.$^{\text{src}\cap\text{tgt}}$ | 1.53 | | 0.55 | | 1.78 | |
| Avg. SAS | 0.73 | | 0.29 | | 0.30 | |

Table 7: Data statistics for the three TST tasks in this work: Sentiment, Formality, and Political Stance Transfer. We compare the S-Emb. + SAS aligned data (the upper part), with the human-annotated reference file (the bottom part). Notice that after alignment, the SAS of training data becomes much closer to that of human reference, which demonstrates we successfully align the non-parallel data into parallel form (comparable to the human reference). Among the three TST datasets, we find the political stance TST has the longest sentence length (see Avg. Sent Len), and more scene entities involved on average (see Avg. # Scene Ent. / Sent). Human written references also reveal similar statistics (see the lower part of the table), which will potentially increase the difficulty for entity-matched transfer. We thus consider the political stance TST as a more challenging task than the other two.

---

**Algorithm 2:** DP-based Best Distillation Strategy Search

---

**Input:** stance saliency $v_i$, distill cost $w_i$, and current cost limit $W_{\text{distill}}$.
Record the number of stance saliency $M = |v_i|$;
Init DP-table $T[M+1][W_{\text{distill}}+1]$ with all 0;
**for** $i = 1, 2, \ldots, M$ **do**
 **for** $j = 1, 2, \ldots, W_{distill}$ **do**
  **if** $j - w_{i-1} < 0$ **then**
   $T[i][j] = T[i-1][j]$;
   Record distillation picking choice $\phi(x_i)$;
  **else**
   $T[i][j] = \max(T[i-1][j],$
   $T[i-1][j-w_{i-1}] + v_{i-1})$;
   Record distillation picking choice $\phi(x_i)$;
  **end**
 **end**
**end**
$\{\phi(x_i)_{i=1}^M\} \leftarrow$ backtracking via records;
**return** best distillation strategy $\{\phi(x_i)_{i=1}^M\}$;

---

scene entities using the method by Wu et al. (2019) (Figure 4 (a)), and based on these entities we compute two distances $d_{\text{Order}}$ and $d_{\text{Exist}}$ as sub-components of $d_{\text{PSV}}$.

For $d_{\text{Order}}$, as shown in Figure 4 (b), we compare each of the $s^{\text{tgt}}$ entities with the $s^{\text{src}}$ entities, and measure how much $s^{\text{tgt}}$ needs to be edited to make the two sets of entities from opposite styles identical to each other. As mentioned in the paper, we use the Levenshtein distance as the edit distance; whose set of edits operation is: 1) insertion of a single token, 2) deletion of a single token, and 3) substitution of a single token. Each of these operations has unit cost, so the Levenshtein

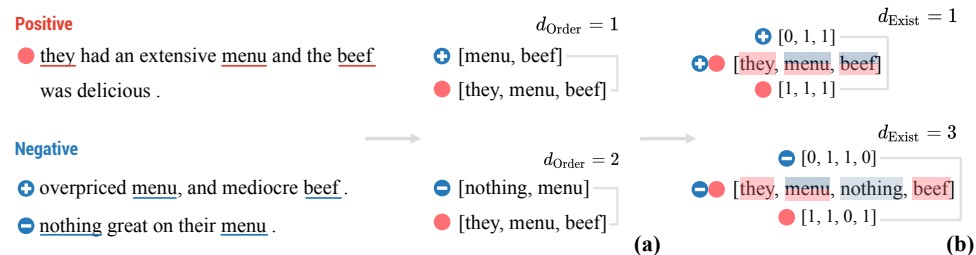

Figure 4: (a) extraction of scene entities (underlined) from source sentence and parallel target sentences. Sentences with a + correspond to $s^{\text{tgt+}}$, and those with a - correspond to $s^{\text{tgt-}}$ (i.e., expert and amateur demonstrations for imitation learning); (b) entity appearing order ($d_{\text{Order}}$, computed by edit distance); and (c) entity existence ($d_{\text{Exist}}$, computed by hamming distance).

distance is equal to the minimum number of operations required to transform target style entities to source style entities.

As shown in Figure 4 (c), we calculate another distance component, $d_{\text{Exist}}$, via hamming distance to emphasize the the difference between the sets of entities in source and target sentences. We first get the union of the entities from a pair of source and target sentences; based on the union set, we can obtain the corresponding binary vector representations of both style entities: 1 means the entity exists in a certain style sentence, and 0 otherwise. Note the binary vectors for each of the sentences will be the same length (since it corresponds to the union of entities). Hamming distance measures the difference between source and target sentences on the shared entities set (this was shown to be an upper bound on the Levenshtein distance). Though there are overlaps between this metric and $d_{\text{Order}}$, we use this metric to *emphasize* the difference in scene entities.

## A.5 MORE DETAILS ABOUT POLICY GRADIENT

We detail our policy gradient algorithm in Algorithm 3. Note that $J_{\text{IL}}^{\text{safe}}$ is a safe-reward threshold. If the greedy policy can already reach this threshold the policy will not be updated. Empirically we set $J_{\text{IL}}^{\text{safe}}$ to $\{0.8, 0.6, 0.4\}$ for the three TST tasks (sentiment, formality, and political stance).

---

**Algorithm 3:** Imitating Learning of LaMer

**Input:** current policy $\pi_\theta$, expert policy trajectory $s^{\text{tgt+}}$, and amateur policy trajectory $s^{\text{tgt-}}$
**Output:** a learned policy $\pi_\theta$ approaching to expert policy $\pi^+$

▷ Get baseline via Greedy Decoding
$\hat{s}_{\text{greedy}} \leftarrow \arg\max \hat{\pi} \sim \text{LM}(a_t|y_{1:t-1})_{t \in [1,T]}$;
Compute $J_{\text{IL}}^{\text{greedy}}$ with $\hat{s}_{\text{greedy}}$, $s^{\text{tgt+}}$, and $s^{\text{tgt-}}$ by Eq. 3;

**if** $J_{IL}^{greedy} \leq J_{IL}^{safe}$ **then**
  ▷ No Optimization on Already Good Policies
  Skip current step Imitation Learning;

▷ Exploration by Sampling Decoding
**while** $i \leq$ max exploration steps **do**
  $\hat{s}_{\text{sample}}^{(i)} \leftarrow$ sampling $\hat{\pi} \sim \text{LM}(a_t|y_{1:t-1})_{t \in [1,T]}$ ;
  Compute $J_{\text{IL}}^{\text{sample}}$ with $\hat{s}_{\text{sample}}^{(i)}$, $s^{\text{tgt+}}$, and $s^{\text{tgt-}}$ by Eq. 3;

▷ Optimize Policy by REINFORCE
Compute policy gradient as $\nabla_\theta J(\theta) = -\mathbb{E}_{\pi_\theta}\left[\nabla_\theta \log \pi_\theta \cdot (J_{\text{IL}}^{\text{sample}} - J_{\text{IL}}^{\text{greedy}})\right]$;
Update policy with policy gradient by taking $K$ steps SGD (via Adam);
**return** $\pi_\theta \rightarrow \pi^+$

---

We choose the REINFORCE algorithm (Williams, 1992) to optimize the current policy $\pi_\theta$. One known issue of the vanilla REINFORCE method is the instability of its optimization because of the high variance of the gradients. A common remedy is to add a baseline function (i.e., a parameterized value-network whose purpose is to estimate the unbiased future behavior of current policy (Ranzato et al., 2016)). Rennie et al. (2017) introduce a better approach that estimates the baseline at test time. They use the greedy decoding results as the baseline in their *self-critic* sequence training algorithm. We adopt their method to provide the baseline, and add an exploration step using sampling decoding. As shown in Algorithm 3, by iteratively switching between these two modes, we calculate corresponding contrastive losses defined by Equation 3, and take the difference of two modes losses as the advantage, feeding into the REINFORCE algorithm to optimize our policy $\pi_\theta$. Finally, the output policy $\pi_\theta$ can generate style transferred sequences resemble the expert demonstrations $s^{\text{tgt+}}$.

## A.6    ADDITIONAL DETAILS ABOUT HUMAN EVALUATION

We provide more details about our human evaluation in this section. The results on sentiment TST are shown in Table 8 and those of formality TST are shown in Table 9.

**Details About Participants and Compensation.**    Participants ($N$=212) were all from the United States and above 18 years old. Participants were required to have a HIT approval rate greater than 95%. The average age of the participants was 34.95 years-old (SD=9.90, Median=32). 141 of the participants (66.5%) were male, and 71 female (33.5%). Participants received 14.41 years of education on average (SD=4.46, Median=16). When asked to self-report their party affiliation, 52.4% participants self-reported as Democratic, 35.3% participants were self-reported Republicans, and 12.3% participants identified as independent. Each participant was compensated 1 USD per task. The average completion time per questionnaire was about 7.6 minutes as reported by the platform, which results in a \$7.9/hr wage, above the regional and federal minimum wage.

Table 8: Human judgement results (on a 7-point scale) of seven existing methods and LaMer on sentiment transfer task. We **bold** the highest average rating. SD: Standard Deviation.

| ■ Sentiment | Style | | Content | | Readability | |
|---|---|---|---|---|---|---|
| Methods | Mean | SD | Mean | SD | Mean | SD |
| CAE (Shen et al., 2017) | 5.09 | 1.49 | 4.79 | 1.56 | 4.84 | 1.50 |
| DelRetrGen (Li et al., 2018a) | 4.94 | 1.62 | 4.73 | 1.63 | 5.05 | 1.39 |
| Dual RL (Luo et al., 2019) | 5.43 | 1.40 | 4.89 | 1.73 | 5.10 | 1.40 |
| Style Transformer (Dai et al., 2019) | 5.09 | 1.55 | 4.71 | 1.56 | 5.04 | 1.50 |
| Deep Latent Seq (He et al., 2020) | 4.89 | 1.79 | 4.63 | 1.76 | 5.05 | 1.49 |
| TSF-DelRetGen (Sudhakar et al., 2019) | 4.66 | 1.35 | 4.73 | 1.24 | 5.30 | 1.13 |
| STARP (Krishna et al., 2020) | **5.50** | 1.20 | 4.65 | 1.40 | 5.33 | 1.20 |
| **Ours**: LaMer | 5.45 | 1.21 | **4.91** | 1.71 | **5.43** | 1.18 |

Table 9: Human judgement results (on a 7-point scale) of seven existing methods and LaMer on formality transfer task. We **bold** the highest average rating. SD: Standard Deviation.

| ■ Formality | Style | | Content | | Readability | |
|---|---|---|---|---|---|---|
| Methods | Mean | SD | Mean | SD | Mean | SD |
| CAE (Shen et al., 2017) | 5.17 | 1.59 | 5.09 | 1.67 | 4.89 | 1.64 |
| DelRetrGen (Li et al., 2018a) | 5.19 | 1.57 | 4.91 | 1.65 | 4.86 | 1.62 |
| Dual RL (Luo et al., 2019) | 5.49 | 1.27 | 5.13 | 1.29 | 5.36 | 1.31 |
| Style Transformer (Dai et al., 2019) | 5.25 | 1.38 | 5.22 | 1.37 | 5.26 | 1.37 |
| Deep Latent Seq (He et al., 2020) | 5.22 | 1.52 | 5.30 | 1.46 | 5.17 | 1.47 |
| TSF-DelRetGen (Sudhakar et al., 2019) | 4.86 | 1.25 | 5.23 | 1.24 | 5.20 | 1.49 |
| STARP (Krishna et al., 2020) | 5.33 | 1.21 | 4.65 | 1.37 | 5.39 | 1.20 |
| **Ours**: LaMer | **5.50** | 1.23 | 5.45 | 1.29 | **5.39** | 1.21 |

**Details About Human Evaluation Procedure.**    We evaluate the style transferred sentences generated by baselines and our LaMer. The sentences are picked from the test set generations for the three

tasks. Each participant was asked to read two sets of sentences (one set containing a source style sentence, corresponding human-written target style sentence as ground-truth, and eight versions of target style sentences generated by seven existing methods and LaMer for comparison). After reading each set, they were asked to rate how well the generated sentences did in style control, readability, and content preservation on a 7-point Likert scales. For example, for content preservation, they were asked *"How much do you agree with the following statement (1- Strongly disagree to 7- Strongly agree): The machine-generated target style text preserves the context of the source style text well (i.e. they talk about the same thing)."*. Higher ratings correspond to better performance.

### A.7 Perplexity measure of LaMer and other baselines

In Table 10 we present the perplexity measure of LaMer and other baselines as a proxy for fluency. As can be seen, certain methods, such as CAE and Style Transformer, have very high perplexity in more challenging TST tasks (e.g., Political Stance), potentially because the task requires longer generation and more entities involved in the sentences (see Table 7 for further discussions). However, it should also be noted that prior work have shown perplexity to be a poor measure of fluency (Krishna et al., 2020; Mir et al., 2019; Lee et al., 2021).

Table 10: Perplexity measure of LaMer and other baselines. We use GPT2-large as the judgement LM, and **bold** the lowest perplexity (the best), and underline the second best.

|  | **Sentiment** | **Formality** | **Political Stance** |
|---|---|---|---|
| **Existing Methods** | PPL | PPL | PPL |
| Input Copy (*ref.*) | 54.0 | 16.2 | 28.0 |
| CAE | 53.4 | 135.4 | 1869.0 |
| DelRetrGen | 18.5 | 30.3 | 86.0 |
| Dual RL | 17.0 | 31.9 | 141.2 |
| Style Transformer | 78.0 | 129.1 | 350.0 |
| Deep Latent Seq | 27.8 | 17.2 | 113.7 |
| **LM-based** | | | |
| TSF-DelRetrGen | 52.1 | 24.5 | 102.3 |
| IMaT | 16.7 | 25.1 | 67.2 |
| START | 12.5 | 15.9 | 33.7 |
| **Ours: LaMer** | | | |
| w/. RD | 13.9 | 13.6 | **32.3** |
| w/. S-Emb. | **11.4** | 13.5 | 47.6 |
| w/. S-Emb. + SAS | 13.3 | **11.7** | 33.5 |

