# OpenReview forum: "Non-Parallel Text Style Transfer with Self-Parallel Supervision"
_ICLR.cc/2022/Conference — ICLR 2022 Poster_

### Official Review · Reviewer_itTT · 2021-10-30

**Correctness:** 4
**Technical Novelty And Significance:** 3
**Empirical Novelty And Significance:** 3
**Recommendation:** 8
**Confidence:** 4

**Main Review:**

*Strengths*

1. I liked the ideas in the paper. As far as I know, scene graphs have not been used previously in style transfer and are a nice way to ensure semantic preservation. I believe parallel data mining has been done before in unsupervised machine translation [1] and pretraining [2] (papers I suggest the authors cite), but not for style transfer. The part of the approach using imitation learning to refine the pseudo parallel data is also new and effective as confirmed by the ablation studies.

2. The authors collect a new dataset for political stance rewriting. This dataset is fairly large (56K pairs) and the task sounds harder than sentiment transfer. This is a valuable contribution and it could be a useful benchmark for future style transfer research. The authors have also carefully curated the dataset to remove hate speech, and used a filtering method (Appendix B) to ensure mined sentences are faithful to the attribute.

3. The authors show improvements in three different text rewriting tasks over several baselines on both automatic metrics and human evaluations. Additionally, the authors provide ablation studies and analyze the choice of their data mining hyperparameters.

*Weaknesses*

1. While I liked the ideas in the paper, I thought the main technical contribution of mining psuedo parallel data from an unpaired corpus is fundamentally limited. First, there's no guarantee the unpaired corpus will have a similar content distribution across attributes (which is required to retrieve roughly parallel sentences). The formality / political stance transfer benchmarks tested were originally parallel datasets, which ensures a shared content distribution. Second, you might need a very large unpaired corpus to ensure this mining is successful, which may not be practical in lower resource settings (this is why recent work is moving towards few-shot setups for style transfer [4, 5]). I suggest the authors try style transfer tasks where the content distributions will be very different across attributes, such as Shakespeare <---> Tweets from [6].

2. Not a very big concern (since your method outperforms baselines on all three metrics), but it will be nice to see an aggregated overall score for style transfer performance (especially on human evaluation). The three style transfer metrics (accuracy, similarity, fluency) often conflict with each other [7], so an overall score is generally more informative. You could do this for both automatic and human evaluations. For human evaluations, since you use 7-point Likert scales you could follow the approach in [8] who calculate the number of instances which got a Likert score of (let's say 5 or more) on all three metrics *simultaneously*. A more general alternative is presented in [4, 6] which can be used for both automatic and human evaluations.

**Minor**

1. The acronym LaMer is not specific to the contributions of this paper (pseudo parallel data, scene graphs, imitation learning). It is also a slang for "dull" in informal settings. I suggest changing this acronym to something else.

2. In 2.4, you could alternatively use unlikelihood training [3] on the amateur policies.

3. I suggest reducing focus on the details of the RL methodology and adding more details on the new benchmark dataset proposed in the main body of the paper.

4. Question about human evaluation ---> did you randomly shuffle the order of generations from different systems? That way you would avoid annotator bias from their previous annotations ("the last generation is always best")

5. Several links seem to be broken, especially those to the Appendix.

6. I suggest the authors add more qualitative outputs from their system into the Appendix part of the PDF, rather than in an external attachment.

[1] - https://aclanthology.org/P19-1178
[2] - https://arxiv.org/abs/2006.15020
[3] - https://arxiv.org/abs/1908.04319
[4] - https://arxiv.org/abs/2110.07385
[5] - https://aclanthology.org/2021.acl-long.293
[6] - https://arxiv.org/abs/2010.05700
[7] - https://arxiv.org/abs/1910.03747
[8] - https://aclanthology.org/N18-1169

**Summary Of The Paper:**

Style transfer is a text generation task where sentences need to be rewritten in such a way that a particular target attribute is introduced (like formality), but at the same time the content is approximately preserved. Style transfer is usually researched in unsupervised settings --- researchers assume no access to parallel data of sentences differing only in the target attribute. However, most prior works assume access to a corpus of unpaired sentences in each of the target attributes.

The main technical idea in this paper is searching for (mining) roughly parallel pairs in the unpaired corpora, to create a "pseudo-parallel dataset". The authors use two methods to extract pairs - (1) semantic similarity based on RoBERTa embeddings; (2) similarity of a scene-graph built by using an off-the-shelf parser. These pseudo-parallel data is used to fine-tune a large pretrained language model (BART). To improve performance, the authors utilize imitation learning via REINFORCE. The key idea is checking whether the current greedy sample is closer by a margin to the closest mined parallel sentence (positive example), compared to other mined parallel sentences (negative examples). If not, sampling is used to explore the space and the sampled sequence is rewarded / penalized according to the increase / decrease in margin using REINFORCE.

The authors test their approach on sentiment transfer, formality transfer and politcal stance transfer (which is a new dataset introduced by them), and note improvements over several popular baselines on automatic metrics and human evaluations. The authors also analyze their choice of hyperparameters and conducted several ablation studies to justify their design decisions.

**Summary Of The Review:**

Overall, I'm in favour of acceptance since the paper has several interesting new ideas, a large new dataset for political stance transfer, and automatic + human evaluation comparing the benefits of their approach against baselines. The weakness #1 prevents me from giving the next higher score (of 8).

-----------

**After author response**: I'm pleasantly surprised by the few-shot style transfer results, and commend the authors for their rigorous rebuttal to all reviewers. I think the paper will be stronger with results on a dataset with a different content distribution between the two styles (irrespective of size, such as Shakespeare <---> Tweets). My overall assessment is more like a 7 or 7.5, which means I'll increase my score to 8.

---

> ### Author Response · Authors · 2021-11-17
> **Response to Reviewer itTT**
>
> We thank the reviewer’s thoughtful comments and constructive suggestions. We are glad that reviewer found strengths in our paper’s novelty, experimental procedure, and contribution to the community. Below we answer your questions:
>
> - **Few-shot cases, and scope of the work.** Thank you for your notes. To explore the few-shot ability of LaMer, we conduct additional experiments in extreme data scarce cases where only 1% of training data is available to train the whole system. As shown in the newly added Table 3, we find LaMer performs well compared with other baselines. The reason could be that the other baselines (except zero-shot GPT-3) all require external systems trained on ample data. For example, TSF-DelRetrGen requires training a classifier to locate style-carrying spans of text. STRAP fine-tunes a GPT-2 model to learn how to paraphrase. IMaT trains a plain Seq2Seq model from scratch as an LM (not pre-trained). All these methods show limited performance in data hungry cases, partially because the compromised external modules would propagate errors to following parts as they are coupled. LaMer instead, built upon pre-trained LMs, requires neither extra data or additional model training, and thus shows superior performance in few-shot cases. We even consider about the two few-shot specific baselines mentioned in your comments [1, 2]; however, none of them release their official implementation. Of note, more unsupervised data will benefit LaMer, but we claim LaMer can do at least as good as many strong LM-based methods in few-shot scenarios. We also include a zero-shot baseline (GPT-3) for reference. Please see revised section 3.2 for more details about these new experiments.
> For TST tasks that have different forms of language, though SAS alignment of LaMer relies on entity alignment for mining parallel corpora, other alignment mechanisms, such as (multilingual) LM alignment (which is our second proposed alignment method) can potentially be used for transfer cases where entity alignment might not be possible (such as ancient English v.s. modern English [1], or even multilingual text style transfer [4]). In general, though we agree that LaMer has certain limitations, we believe the ideas conveyed in this paper push forward a simple and efficient TST framework without the need for additional training of external models or extra data (or annotation).
>
> - **About better evaluation metrics, and new human evaluation statistics.**
> Thanks for your great suggestions! As shown in revised Table 1, we have replaced perplexity with GM (geometric mean of ACC and BLEU) to measure the overall performance. Also, following your suggestion, we have calculated the ratio of annotators who rated the transfer above a certain point and added this information in Tables 5 and 6.
>
> - **About name: LaMer.** Thank you for your note regarding the name LaMer. We did not change the name in the revision to avoid confusion but we will definitely consider renaming our method for the camera ready (How about NEXUS, which means connection?). :)
>
> - **About presentation.** Thanks for your suggestion. We have moved the details about RL learning to appendix and have fixed the link issues! We have also cited the paper [3] you mentioned in "Strength 1".
>
> - **About human evaluation.** Yes. We design the questionnaire with Qualtrics and add a random sampler for shuffling the sentences. The annotators were not aware of which system generates the sentence.
>
> - **About sample generations.** As noted by the reviewer, we have included a text file with examples of sample generation for reviewing purposes. Based on your suggestions, we will add several samples from the file to our appendix for the camera ready (in addition to sample generations already shown in Table 8).
>
> Thanks again for your thoughtful reviews and helpful suggestions!
>
> [1] [Few-shot Controllable Style Transfer for Low-Resource Settings: A Study in Indian Languages](https://arxiv.org/pdf/2110.07385.pdf)
>
> [2] [TextSETTR: Few-Shot Text Style Extraction and Tunable Targeted Restyling](https://aclanthology.org/2021.acl-long.293.pdf)
>
> [3] [Pre-training via Paraphrasing](https://arxiv.org/pdf/2006.15020.pdf)
>
> [4] [Paraphrasing for Style](https://aclanthology.org/C12-1177)

---

> > ### Comment · Reviewer_itTT · 2021-11-23
> > **Thank you for your detailed response, one clarification question**
> >
> > Thank you for your detailed response! I had one clarification question about the few-shot experiments. There are two ways to construct a 1% training dataset:
> >
> > 1. Take 1% of the unsupervised training corpus (1% for each style). Perform mining on it. Use all mined pairs to train the model.
> > 2. Perform mining on the entire unsupervised training corpus. Take 1% of the mined pairs for training the model.
> >
> > Which one of the two methods are you using for your few-shot setting?

---

> > > ### Author Response · Authors · 2021-11-23
> > > **Further response to reviewer itTT**
> > >
> > > We used the first method. We ran the entire system with the limited data (1% of the 52k formality TST training data), including the mining stage. Mining on this produces around 520 $s^{\textrm{src}}$, and each $s^{\textrm{src}}$ has multiple $s^{\textrm{tgt}}$ as demonstrations --- the number of $s^{\textrm{tgt}}$ depends on the top-$p$ and -$k$ you set for alignment (the impact of these parameters is shown in Figure 3. We used $k=200, p=0.4$ for formality TST). We then split these one-to-N $s^{\textrm{src}}$- $s^{\textrm{tgt}}$ pairs to one-to-one pairs (around 1800) for MLE training, and use the one-on-N version (around 520) for IL refinement.
> > > For other baselines we used the same size of data (i.e.,1% of the 52k formality TST training data), which was used to train the core components of those baselines, such as the style classifier used by TSF-DelRetrGen.
> > >
> > > We hope this clarifies our approach to the few-shot setting.

---

> > > > ### Comment · Reviewer_itTT · 2021-11-23
> > > > **Thanks for your response, a quick follow-up question**
> > > >
> > > > Thanks for your reply! As a quick clarification question, what do you mean by "each s^src has multiple s^tgt as demonstrations?" Do you mine demonstrations from a pool of only from 1% of the D_tgt corpus, or the entire D_tgt?

---

> > > > > ### Author Response · Authors · 2021-11-23
> > > > > **Further clarification**
> > > > >
> > > > > The demonstrations come only from the pool of 1% of the $D^{\textrm{tgt}}$. Specifically, the whole procedure is as follows:
> > > > >
> > > > > 1. We sample 1% of the data points from each of the formal and informal sentences (which corresponds to around 520 formal and 520 informal sentences). Now we have the new $D^{\textrm{src}}$ (formal) and $D^{\textrm{tgt}}$ (informal); each with a size of 520.
> > > > > 2. We run alignment on $D^{\textrm{src}}$ and $D^{\textrm{tgt}}$. Each $s^{\textrm{src}} \in D^{\textrm{src}}$ will be mapped to multiple $s^{\textrm{tgt}} \in D^{\textrm{tgt}}$ by the alignment algorithm. The number of these one-to-N pairs is 520 since we have 520 $s^{\textrm{src}} \in D^{\textrm{src}}$.
> > > > > 3. We then split these one-to-N pairs into one-to-one pairs, resulting in 1800 $s^{\textrm{src}} - s^{\textrm{tgt}}$ pairs.
> > > > > 4. We use the one-to-one pairs for MLE training, and the one-to-N pairs for IL refinement.
> > > > >
> > > > > Thanks again and please let us know if you have further questions!

---

> > ### Comment · Reviewer_itTT · 2021-11-24
> > **Thank you for the detailed rebuttal, raised my score to 8**
> >
> > I'm pleasantly surprised by the few-shot style transfer results, and commend the authors for their rigorous rebuttal to all reviewers. I think the paper will be stronger with results on a dataset with a different content distribution between the two styles (irrespective of size, such as Shakespeare <---> Tweets). I've increased my score to 8 (I've updated main review).
> >
> > Do you have a sense of why the approach works reasonably in few-shot settings? Is it the case that you have sufficient entity alignments / syntactic alignments (even if there aren't enough full semantic alignments)? Is hallucination more common in the few-shot setting?
> >
> > NEXUS is certainly a better name, assuming it's an acronym for something related to the technical contribution in this work.

---

> > > ### Author Response · Authors · 2021-11-24
> > > **Thank you, and answers to your question**
> > >
> > > Thank you for your response and we are grateful that you have updated your recommendation.
> > >
> > > 1. Thank you for the suggestion regarding experiments on an additional dataset with different content distribution. We agree that such an experiment would strengthen our paper. We plan to run such an experiment on at least one dataset and include the results in the camera ready.
> > >
> > > 2. We believe our method works reasonably well in few-shot settings mainly because of the imitation learning (IL) refinement. Specifically, though the aligned pairs could be fewer in few-shot settings, the IL refinement will encourage the model to focus more on the really good parallels as the positive demonstration and pay less attention to the bad ones as the negative demonstrations (though there will of course be fewer negative demonstrations in few-shot settings).
> > >
> > > We thank the reviewer again for the thoughtful reviews and great suggestions!

---

### Official Review · Reviewer_Hwf9 · 2021-10-30

**Correctness:** 3
**Technical Novelty And Significance:** 2
**Empirical Novelty And Significance:** 3
**Recommendation:** 6
**Confidence:** 3

**Main Review:**

Strengths:

- The paper uses a novel combination of existing approaches to achieving competitive results.

- The code provided with the paper is clean, structured, and can be useful for developing future models for the same task.

- The paper itself is well written and structured.

Weaknesses and questions:

- The authors claim "existing methods learn a mapping without considering the self-parallelism, ... they tend to learn the mapping between source and target style by randomly mapping sentence pairs." However, no proofs or references for such behavior were given. On the contrary, most of the works mentioned as related have no such problematic behavior by design.

- The first step of the proposed approach (matching sentences from the source and target corpora) implies that both corpora have a similar distribution. However, it is not the case for some definitions of "styles." This assumption possibly limits the method's applicability, but it wasn't addressed in the motivation or discussion sections.

- Moreover, the token-level scene preservation scheme implies that scene entities couldn't be style markers themselves. This assumption also limits possible applicability until the opposite is proven.

- Again, the part of the loss is based on the difference in the orders of scene entities, but in practice, it can be an important part of the style. Consider the example from the 'political stance' dataset presented with the paper:
Style1: Mulvaney tapped to lead Trump s budget office.
Style2: Trump picks debt warrior Mulvaney to lead White House budget office.

- There is a known inevitable trade-off between content-preservation and style-transfer metrics. Thus, to show the improvement of the new model, one needs to demonstrate Pareto-improvement both of them (in this case, this means the improvement both in terms of ACC and BLEU metrics). However, the reported results show the TSF-DelRetGenLM model is better in terms of one of these two metrics for both sentiment and formality datasets. Thus, the evaluation shows the comparability of the model with the baselines on two standard datasets (and the superiority only on the novel proposed dataset). At the same time, the authors claim "SOTA results according to many metrics on the three TST tasks," which doesn't sound fair.






**Summary Of The Paper:**

The paper addresses a text-style transfer task based on non-parallel datasets.

The authors propose a three-step approach:
- First, to match each source sentence with several sentences from the target style dataset using sentence embedding similarity and scene graph matching.
- Next, to fine-tune a seq2seq network (BART) in an NMT-like setup with several references.
- Finally, to use imitation learning to enforce the loss contrast between the best target candidate and all others.

The authors propose a new dataset for the TST task, called "political stance transfer," and use it for evaluation among the previously known datasets. The results of such a combined approach are competitive with previous works.

**Summary Of The Review:**

I like the proposed approach and the clarity of the paper and code.
I'm slightly concerned with the paper's novelty since it presents a combination of previously known methods, but still, I think it can be useful for the community.
The main concern is somehow ambiguous motivation and definition of the scope of applicability of the proposed approach (check the questions above).
I believe this should be clarified.

My score for the paper is "marginally above the acceptance threshold".

---

> ### Author Response · Authors · 2021-11-17
> **Response to Reviewer Hwf9**
>
> We thank the reviewer’s thoughtful comments and suggestions, and we are glad that you found our idea interesting and novel, and could be useful for the community. Below we answer your questions:
>
> - **Regarding related work.** Thank you for pointing this out. We indeed find that most non-LM based methods do not consider self-parallelism within the data by just taking the unsupervised data as the input without any alignment. While some recent works do seem to notice the inherent self-parallelism and propose several strategies to leverage this. LaMer differs from them as it requires neither extra data (such as paraphrases data used by STRAP) nor additional training on external systems (such as style locator used by TSF-DelRetrGen). We have added additional references next to our claims in the revised introduction.
>
> - **Regarding scope of LaMer.** We also added additional discussion in section 2.2 regarding the scope of our SAS alignment and describe how other non-attribute-based alignment methods (such as multilingual LM alignment) can be used for TST scenarios where entities may not be shared across styles (e.g., multilingual TST). In general, though we agree that LaMer has certain limitations, we believe the ideas conveyed in this paper push forward a simple and efficient TST framework without the need for additional training of external models or extra data (or annotation).
>
> - **Regarding scene entities.** We agree that the scene entity can sometimes carry style; however, note that our scene graph alignment only provides weakly parallel samples, the exact mapping relationship is learned by the text2text LM, which means that aligned entities are not a *hard* constraint for learning but provide *soft supervision*. Our experiments demonstrate that learning over such samples can empirically lead to good TST performance.
>
> - **Regarding the token-level scene preservation.** $d_{\textrm{Order}}$ and $d_{\textrm{Exist}}$ are actually taken into consideration with different weights. For the political stance TST we assign little weight to the order as the sentences are longer and more complicated. We have updated the equation in Token-level Scene Preservation of section 2.3 to show how we sum up the two weighted distances.
>
> - **About SotA claims.** In the revised Table 1, we have replaced perplexity with GM (geometric mean of ACC and BLEU) to measure the overall performance. We also made sure to not claim SotA results on all metrics for all tasks. Note that after closer examination, we find the BLEU score calculation script used by TSF-DelRetrGen is slightly different from other methods: most existing methods (including ours) measures the average n-gram (n ranges from 1 to 4) matching (e.g., [style transformer](https://github.com/fastnlp/style-transformer/blob/master/evaluator/evaluator.py) uses sentence BLEU, which assigns equal weights for 1-4 gram matching). TSF-DelRetrGen, instead, uses the **max** n-gram (n ranges from 1 to 4) matching to compute BLEU (see line 14 in their [evaluation code](https://github.com/agaralabs/transformer-drg-style-transfer/blob/master/evaluation_scripts/bleu.py)), which could partially explain its relatively high BLEU scores. We re-evaluate their results with the standard scripts to align with other methods.
>
> Thanks again for your thoughtful reviews and helpful suggestions! Hope our revision can solve your concerns and questions!

---

> > ### Comment · Reviewer_Hwf9 · 2021-11-21
> > **Thank you for the quick and detailed update**
> >
> > Thanks for the quick and detailed update. Part of my concerns was successfully solved; however, I still have some:
> > * "**We also made sure to not claim SotA results on all metrics for all tasks**," but you still claim "LaMer has the best overall performance," which should be supported by more solid evaluations, in my opinion.
> > * "**we have replaced perplexity with GM (geometric mean of ACC and BLEU) to measure the overall performance**." I don't think it is the right thing to do: while perplexity could be useful as a proxy for fluency, G-score (or H-score) can't be claimed as a universal single number metric without some additional proof and research on the right way to weight its components. The given references lack such explanations or proofs, so the use of G-score is unmotivated here.
> > * "**d_order and d_exist are actually taken into consideration with different weights. For the political stance TST we assign little weight to the order as the sentences are longer and more complicated**" -- this raises the question of how to pick the correct value for this new hyperparameter.

---

> > > ### Author Response · Authors · 2021-11-22
> > > **Further response to reviewer Hwf9**
> > >
> > > Thank you for the response. We are happy our revision and previous response addressed parts of your concerns. We have addressed the new issues raised below and included a revised version of the paper.
> > >
> > > #### **SotA Claims**
> > >
> > > We have replaced “best overall performance” to “competitive performance” and have specifically mentioned the metrics in which LaMer outperforms other methods (see Section 3.2 in the revised version). Thank you for pointing this out.
> > >
> > > #### **Performance Metrics**
> > >
> > > We have reintroduced perplexity in Section 3.2 with a reference to a new table in the appendix showing our original results (Table 12 in A.9). We decided to remove perplexity from the main table as other reviewers pointed out recent work that shows its shortcomings. For example, [1, 6, 7] find perplexity is actually a poor measure for fluency because “1) it is unbounded and 2) unnatural sentences with common words tend to have low perplexity.” We believe our human evaluations shown in Table 6 (in the main paper) and Tables 9, and 10 (in the Appendix) are a better measure of fluency/readability since they measure it directly, whereas perplexity is a proxy for fluency/readability. However, if the reviewer feels strongly about the inclusion of perplexity in the main paper, we would be happy to do so in Table 1.
> > >
> > >
> > > Informed by several prior work, we believe that GM can serve as a single score with which readers can compare different TST methods easily (though we agree with the reviewer that GM is not a perfect aggregate score). For instance, GM (i.e., G-score) as an aggregate score is used in the following papers (we have cited these papers as the motivation for use of GM in Section 3.2):
> > >
> > > - Reformulating Unsupervised Style Transfer as Paraphrase Generation [[1]](https://aclanthology.org/2020.emnlp-main.55.pdf) (see Table 1)
> > > - Unpaired Sentiment-to-Sentiment Translation: A Cycled Reinforcement Learning Approach [[2]](https://aclanthology.org/P18-1090.pdf) (see Tables 1,2, and 4)
> > > - Domain Adaptive Text Style Transfer [[3]](https://aclanthology.org/D19-1325.pdf) (see Tables 2 and 6)
> > > - A Dual Reinforcement Learning Framework for Unsupervised Text Style Transfer [[4]](https://www.ijcai.org/proceedings/2019/0711.pdf) (see Tables 1 and 3)
> > >
> > > Additionally, our results in Figure 3 show the trade-off between ACC and BLEU when running the model under different hyperparameters (the $k$ and $p$ in our case). Finally, as suggested by other reviewers, in the revisions we also report additional metrics, such as the $J$-score when we evaluate LaMer in few-show (and full-data) scenarios (see Table 2), which is introduced in [1] as a more advanced overall metric.
> > >
> > > We hope our clarifications and revisions have addressed your concerns regarding the evaluation metrics.
> > >
> > >
> > > #### **Picking the weights for $d_{\textrm{Order}}$ and $d_{\textrm{Exist}}$**
> > >
> > > We pick the weights by empirical observation. We run repeated experiments ranging the $\alpha$ from 0 to 1 by step-size of 0.1, and pick the best $\alpha$ that can lead to the best performance with respect to GM. We have added this explanation to footnote 2 in Section 2.4 (page 5).
> > >
> > >
> > > We once again thank you for your time and detailed comments, and we believe your suggestions have made our paper much stronger!
> > >
> > > ---
> > > References:
> > >
> > > [1] [Reformulating Unsupervised Style Transfer as Paraphrase Generation](https://aclanthology.org/2020.emnlp-main.55.pdf)
> > >
> > > [2] [Unpaired Sentiment-to-Sentiment Translation: A Cycled Reinforcement Learning Approach](https://aclanthology.org/P18-1090.pdf)
> > >
> > > [3] [Domain Adaptive Text Style Transfer](https://aclanthology.org/D19-1325.pdf)
> > >
> > > [4] [A Dual Reinforcement Learning Framework for Unsupervised Text Style Transfer](https://www.ijcai.org/proceedings/2019/0711.pdf)
> > >
> > > [5] [Enhancing Content Preservation in Text Style Transfer Using Reverse Attention and Conditional Layer Normalization](https://aclanthology.org/2021.acl-long.8.pdf)
> > >
> > > [6] [Evaluating Style Transfer for Text](https://aclanthology.org/N19-1049/)
> > >
> > > [7] [Text Generation by Learning from Demonstrations](https://openreview.net/pdf?id=RovX-uQ1Hua)

---

> > > > ### Comment · Reviewer_Hwf9 · 2021-11-22
> > > > **Still no known proof for GM as a good metric**
> > > >
> > > > > "Informed by several prior work, we believe that GM can serve as a single score with which readers can compare different TST methods easily"
> > > >
> > > > [1], [3], [4] used references to the [2] as a support for using GM.
> > > >
> > > > [2] provides one very weak comment on it, literally: **"The G-score is one of the most commonly used "single number" measures in Information Retrieval, Natural Language Processing, and Machine Learning."**
> > > >
> > > > I strongly advise against using unchecked and unsupported by research metrics as support of your results.

---

> > > > > ### Author Response · Authors · 2021-11-22
> > > > > **We agree and have updated our paper accordingly**
> > > > >
> > > > > We thank the reviewer for their fast response (especially on a Monday), allowing us to  comment before the end of the discussion period. On further deliberation on the reviewer’s point, we agree that the use of GM/G-score seems to be based on one paper which then propagated to other papers without much support. Since several of the baselines we are comparing against (DualRL and STRAP) use the GM we will keep it for ease of comparison. However, we have removed the emphasis on it as the one aggregate metric to be looked at. We hope that our comprehensive list of metrics (ACC, BLEU, $i$-PINC,GM, $J$-score, and perplexity), all of which are used by at least one of our baselines, can provide a better picture of our method’s performance with respect to other methods. We made sure that the metrics of all of our baselines are represented in our analysis for more direct comparison.
> > > > >
> > > > > In the same way, we have also added a sentence in Section 3.2 to emphasize that GM is only included for comparison purposes and that it is not a well-studied metric for TST.
> > > > >
> > > > > We once again thank the reviewer for their time and valuable comments that have strengthened our paper.

---

### Official Review · Reviewer_8fDY · 2021-11-02

**Correctness:** 3
**Technical Novelty And Significance:** 3
**Empirical Novelty And Significance:** 3
**Recommendation:** 8
**Confidence:** 4

**Main Review:**

**Strengths**

The paper tackles an important issue of text style transfer with novel ideas that aims to preserve content in an effective manner.
The paper is relatively well written with justifications and reasons for the choices. The authors could have given examples
The paper presents a simple idea to use reinforcement learning: imitation learning in particular  to better maintain the content between the two styles.

**Weaknesses**

1. The authors assume that  parallel sentences inherently exist between the source and the target style corpus. However, the existing unsupervised text style transfer methods do not make such assumptions. Since the entire basis of the paper is on such an assumption, providing examples of the inherent parallel sentences is crucial. The paper can be made stronger by  providing  examples (either manually picked or mined).

2. **Scene graph alignment** - What are the scene entities? The authors mention that the use (Wu et al. 2019) paper on Unified Visual Semantic Embeddings. The reviewer skimmed through the other paper as well. However, which parser is used in the current paper?  Wu et al. use a syntactic parser to mine the objects in the sentence, adjectives, nouns, subjects of verbs etc and initialize an embedding for different entities and learn a joint embedding space. Here are my concerns with respect to this.
a) The authors need to be clear on the parser which is used to parse the sentence in this paper. Referring to Wu et al.’s paper without describing what the scene graph entities make it unclear on the reason to include this as a refinement step.
b) If the above parser to extract objects is used, how is it important for style transfer? For example, the style transfer datasets used in the paper might not share any “visual” entity like a clock, plate etc. Instead they might share some abstract entities like the service provided, the cleanliness of the restaurant etc.

     The reviewer understands that the sentences that talk about similar entities are important, the approach to obtain those entities is unconvincing. Seeing Figure S1 in the appendix, it is clear that the scene entities are words that are salient in the given style or words that carry content. Although the reviewer understands the idea, it would be a good idea to rewrite this to make it clear to the reader.


3. **Evaluation Measures**: The used evaluation measures in the paper have known problems. For example, using BLEU does not measure semantic similarity with the input sentence and certain sentences with unnatural structure can still have low perplexity scores. Refer to Kalpesh Krishna et al 2020(https://arxiv.org/abs/2010.05700), which is included as a baseline method to compare against. The adoption of these measures would have made the paper stronger. Although the reviewer agrees that the majority of the text style transfer papers use the metrics used in this paper, the reviewer would appreciate if the authors could evaluate based on the paper from Krishna et al 2020 and add it in the Appendix or supplementary material

4. **Evaluation**: The evaluation section should concentrate on the major contribution of the paper: maintaining the content between the two styles well. For example, with imitation learning we expect the BLEU metrics (and the i-PINC metric)  to do better because imitation learning aims to preserve the content in a better manner between the source and the target style which needs to be the highlight of the evaluation section. The other metrics might not be the focus of the paper and achieving high measures on them is an added bonus. The word “performance” is excessively used in the evaluation section. Instead, the specific measure that the others are talking about should be mentioned.
	Also, Mean and Standard Deviation with multiple experiment runs are missing from the evaluation section.

5. **Conclusion**: The conclusion is a summary of the main findings of the paper. It would be ideal if the authors can provide implications of the current work going forward. Can this be extended to other works that try to control the fine-grained syntax of the sentence etc would be a welcome addition to the conclusion

6. **New Dataset**: More details on the new dataset introduced in the paper is needed. Why is the new dataset more challenging compared to the other two datasets that are mentioned in the paper?

**Minor Typos, Grammatical Errors and Other Readability Issues**

1. There is an extra space on Page 2 Para 2 Line 2 after Figure till (b)
2. The authors introduce the term “scene graphs” in the introduction on Page 2 Para 2 and Line 4. Readers who are not aware of the computer vision literature might be left wondering on how scene graphs are relevant for Text Style Transfer
3. The rows showing ablations in Table 1 are confusing “w/” is read as without or with. The authors could instead use “+” or “-” symbol to indicate with or without
4. Table  1 highlight boxes need to be explained to the reader.  Please mention that you are highlighting the best performing method for that metric. Also some of highlight boxes are missing in the Sentiment dataset


**Summary Of The Paper:**

The paper proposes to transform the popular unsupervised style transfer to a semi supervised task. First the paper proposes simple methods to mine parallel sentences from two unlabelled corpora. Further, they use the mined parallel sentences, in a MLE based training (sequence to sequence framework using BART) and refine it using Contrastive Learning based imitation learning. The paper achieves improvements on preserving the content between the transferred sentences. In addition, the authors also propose a new dataset for political style transfer in a semi-automated way which is a welcome addition to the paper.


**Summary Of The Review:**

1. Overall, the paper is easy to read and the ideas to improve content preservation between the source and the target sentence is simple and effective. The ablation studies and parameter sensitivity studies are well performed.
2. The reviewer would require more details on the entities that are preserved between the sentences, examples on the inherent parallel sentence between the two corpora in the introduction.
3.  Problematic evaluation measures used in the literature need to be revised
4. The evaluation section needs to focus on the main contribution of the paper. Splitting Section 3.2 into more paragraphs might be a starter.
5. The benchmark table needs to have multiple runs with mean and standard deviation reports.

---

> ### Author Response · Authors · 2021-11-17
> **Response to Reviewer 8fDY**
>
> We thank the reviewer’s thoughtful comments and suggestions, and we are glad that you found our idea interesting and novel. Below we answer your questions:
>
> - **Better motivation and samples.** Thanks for your suggestion. We have added an example of SAS mined sentences with corresponding scene entities annotated in Figure 2 (to qualitatively illustrate our assumption). We further justify our assumption through empirical observation on the data: In Table 7 (located in the appendix) we have shown that if NO SAS alignment happens (RD), there would be huge discrepancy on the average SAS score between training data and human references (e.g., in Formality TST, 0.29 with aligned by SAS and 0.01 without alignment). Since the human references are already parallel style transfer sentences, our success on the alignment demonstrates that there is self-parallelism existing in the unsupervised data. Additional human evaluation confirms that our aligned sentences are indeed similar in style-independent content---please take a look at Table 5 and related discussion.
>
> - **Explanation about scene graph lignment.** We use [the parser released by the author of Wu et al.'s paper](https://github.com/vacancy/SceneGraphParser), which outputs mined scenes graphs in *<subject-relation-object>* triplets. In our case, we only consider the end nodes of these triplets (which are *subj.* and *obj.*) as the “scenes entities”. We have made this clear in our revised version (see section 2.2).
>
> - **Better Evaluation Metrics.** Thanks for your great suggestion! We agree that existing metrics have flaws, so in our revised version, besides the ACC and BLEU in Table 1, following your suggestion, we use the new $J$-score framework proposed by [1] in Table 2 to evaluate the few-shot cases (in addition to the full training data experiments). We also include a GM metric which is the geometric mean of ACC and BLEU to represent the overall performance of TST models. We hope these changes can make the evaluation fairer and easier to interpret.
>
> - **More Concentrated Evaluation.** We have heavily revised our evaluation section with reorganized focus on specific perspectives (see section 3.2). We now present the overall and few-shot performances, ablation study, and human evaluation in order, and include more specific conclusions and discussion. For example, as you suggested, we have revised the ablation study section on imitation learning, and highlighted its effectiveness on content preservation (retitled with “IL refinement is crucial for LaMer content preservation”). We have included the SD of results in Table 2 as well.
>
> - **Extended conclusion.** We have revised our conclusion and now it not only summarises our main takeaways and advantages of LaMer but also provides implications for future research. Please take a look (section 5) and let us know if anything can be improved!
>
> - **Explanation about why political stance TST dataset is challenging.** Thank you for your suggestion. We have included more details about the datasets, and discussed the major differences between political stance transfer and the other two TST datasets in the revised Table 7 caption (located in the appendix). In general we find that political stance TST requires transfer which is much longer and contains more entities, which we believe makes it a more challenging TST task.
>
> - **Minor typos, or presentation improvement.** Thanks for the detailed comments! We have fixed the links to the appendix and several typos. In the revised section 2.2 we have explained what the scenes entities are and how we parse them. We have added a visualization sample of scenes entities in parallel sentences as in Figure 2. We have revised Table 1 for better readability and could take your suggestions in our final version. Thanks again!
>
> Finally, we appreciate the suggestions on the paper presentation. We hope our revised version could solve your concerns. Thanks again for your thoughtful reviews!
>
> [1] [Reformulating Unsupervised Style Transfer as Paraphrase Generation](https://arxiv.org/pdf/2010.05700.pdf)

---

### Official Review · Reviewer_7agN · 2021-11-02

**Correctness:** 3
**Technical Novelty And Significance:** 2
**Empirical Novelty And Significance:** 3
**Recommendation:** 3
**Confidence:** 4

**Main Review:**

Strengths

- A step-wise way of training text style transfer models on non-parallel datasets
- A new dataset for style transfer, collected from the political domain.

Weakness

- First, I am not sure about the novelty of the proposed method. Given the ideas of MLE and RL/IL (specifically, the INFORCE algorithm used in this work) has been used in text generation for a while, I am not sure how much novelty here by simply combining them together.
- Second, the writing of this work needs to be improved.
  - In section 2.2, the Scene Alignment Score seems to be an important component of the proposed first step, however, I don't think this work ever (1) explain what is a scene graph and (2) justify the validity of using the Scene Alignment Score for alignment. The numbers in Table 1 may provide some empirical evidence for the second question, but it is not sufficient.
  - I was confused by the terms used in section 2.4, including "Reinforced Imitation Learning", "Reinforced Policy Gradient". To be specific, based on the description, I didn't understand the difference between reinforced imitation learning and imitation learning. In addition, I am also not sure what reinforced policy gradient is. It looks like the description in section 2.4 mixes many terms together, without an appropriate explanation.
  - In Algorithm 1, what is $J_{IL}^{safe}$?
  - About table 1, I am not sure how the highlights were selected in this table. Apparently, not all the highlights are the best results.

**Summary Of The Paper:**

This paper proposes an approach to addressing text style transfer with non-parallel data. The basic idea is to follow three steps: (1) for a given source sentence, mine some nearly parallel sentences from the target domain; (2) perform an MLE learning; and (3) augmented with an imitation learning.

The proposed method was evaluated on three text style transfer tasks: (1) sentiment transfer; (2) formality transfer; and (3) a new task called political stance transfer.

**Summary Of The Review:**

My major concern of this work is the technical novelty. In addition, the writings of this paper also needs to be improved.

---

> ### Author Response · Authors · 2021-11-17
> **Response to Reviewer 7agN**
>
> We are glad you found our idea novel, and evaluation results mostly convincing. Below we address your questions:
>
> - **Novelty.** The REINFORCE algorithm we use in our work can be viewed as an optimization procedure, while the core idea or novelty of our method is to use scene graphs (an idea originally from CV) to mine demonstrations from the unsupervised data, and use imitation learning with contrastive loss to learn how to transfer style in an efficient way. The major difference (and novelty) between our method and some recent paraphrase-based TST models is that we don’t need extra supervision from externally trained models (such as paraphraser used by STRAP and style locator used by TSF-DelRetrGen), which will bring additional bias to the training. LaMer, instead, uses the *self-supervision* from the unsupervised data itself with contrastive loss, and learns how to transfer in one-step seq2seq training, which shows superior performance across different settings.
>
> - **About scene entities, and additional human evaluation.** Thanks for your suggestions on the presentation of our paper. We have included an example (Figure 2) to show what scene graph entities look like when we mine using the scene graph parser (see footnote 1 for information about the parser). In revised section 2.2 we also explain how we extract scenes entities from the parser's output. We run additional human evaluations to demonstrate that the SAS-mined pairs share similar style-independent content based on human judgement (newly added Table 5). Please take a look and we are happy to hear further suggestions!
>
> - **About Policy-based RL, REINFORCE, Imitation Learning.** Policy gradient is a common optimization method adopted by many policy-based RL algorithms, and REINFORCE is just one of them. Imitation learning is a branch of RL: Instead of learning from the samples derived from the environment, IL learns from expert demonstrations directly (e.g., learning how to self-driving from human drivers' behavior records rather than vehicles' sensor data), so that it can avoid low sampling efficiency problems in RL and can have fast convergence. Following your suggestion, we have changed the subtitle of section 2.4.1 from “Reinforced Policy Gradient” to “Policy Gradient by REINFORCE”. We also moved the detailed learning algorithm to the appendix, and left the core equation (Eq. 4) there with a more easy-to-understand description in the revised paper. We rephrased the discussion about MLE, IL, and reward-based RL as well for easier understanding. Please let us know if any further improvement can be made!
>
>
> - **What is $J_{\textrm{IL}}^{\textrm{safe}}$?** $J_{\textrm{IL}}^{\textrm{safe}}$ is a safe-reward threshold. If the greedy policy can already reach this threshold then the policy will not be updated. Empirically, we set $J_{\textrm{IL}}^{\textrm{safe}}$ to $\{0.8, 0.6, 0.4\}$ for the three TST tasks (sentiment, formality, and political stance), which leads to decent performance. We clarified this in section A.5 of the appendix in the revised paper.
>
> - **Confusion on highlights.** In our revised paper, we have introduced the GM (geometric mean of ACC and BLEU) as an overall performance metric. Now we only color the best results in GM (and $i$-PINC) and underline the second best, to let readers focus on the overall performance in the first part of evaluation. We have made this protocol clear in the caption of Table 1.
>
> Finally, we thank the reviewer for the thoughtful comments and constructive suggestions. We hope our revised paper and additional experiments can provide further clarity!

---

### Official Review · Reviewer_53GA · 2021-11-04

**Correctness:** 3
**Technical Novelty And Significance:** 3
**Empirical Novelty And Significance:** 2
**Recommendation:** 6
**Confidence:** 5

**Main Review:**

Strengths:
1. The scene graph similarity used to select pseudo parallel pairs is novel and makes sense to me.
2. The reinforced imitation learning shows better performance compared with baseline of MLE loss for one-to-many mapping.
3. The empirical results establish the effectiveness of this method.

Weaknesses:
1. This method is based on the assumption that there are parallel pairs in the original corpora of two styles so that sentences of the same content and different styles can be found out, which limits the use of the method to cases where abundant corpora are existing and there are parallel sentences there in the corpora. In those cases where only small unsupervised corpora are there and there are no sentences pairs sharing the same content, this method would not work.
2. In the current used three datasets, I would like to know how good are those paired sentences selected out. This kind of quantitative human evaluation of constructed pseudo-parallel corpora is very important to establish the effectiveness of this work, which cannot be lacking.
3. What is the BLEU score? Is it self-BLEU or ref-BLEU? If ref-BLEU, is it calculated on one human reference or four? I know at least for Sentiment transfer and formality transfer, the test sets contain four references.
4. In Table 1, according to the Tabl1 of the original DualRL paper: https://arxiv.org/pdf/1905.10060.pdf, the BLEU scores can be 55.2 and 44.9 for Yelp and GYAFC datasets, while the scores reported in this work are significantly lower. I am not sure whether the references are different here. Please explain this gap.
5. In Table 1, the performance of Ours w/. RD also looks very high, especially for the Political Stance dataset, the difference between w/. RD and w/. S-Emb are very small for BLEU and PPL of w/. RD is even lower, which contradicts to my intuition. Using a pseudo-parallel corpora constructed by random selection should not produce any good performance, should it? This kind of RD baseline even outperforms many strong SOTA baselines, for example of DUAL RL, which is ironic.
6. In Table 1, it shows that the SAS score is only helpful for one datasets out of three, then why is it so? Why is it only useful in the formality case?
7. This work is inspired by this work: https://aclanthology.org/D19-1306/, however, it has never been compared with it.

**Summary Of The Paper:**

This work has proposed a new method to textual style transfer, which is based on the assumption that there exist some pseudo-parallal sentences pairs between two styles. It first construct synthetic parallel corpora by using two similarity measures: semantic similarity based on larger LMs and scene graph similarity. Then it trains the generation model via reinforced imitation learning.

**Summary Of The Review:**

This work has merits, but its weaknesses are also salient as mentioned above. I am looking forward to author responses for addressing my concerns, or I vote for rejecting this work.

---

> ### Author Response · Authors · 2021-11-17
> **Response to Reviewer 53GA**
>
> Thanks for reviewing our paper, and providing your valuable feedback! Below we address the issues raised:
>
> - **About few-shot cases.** To explore the few-shot ability of LaMer, we conduct additional experiments in extreme data scarce cases where only 1% of training data is available to train the whole system. As shown in the newly added Table 3, we find LaMer performs well compared with other baselines. The reason could be that the other baselines (except zero-shot GPT-3) all require external systems trained on ample data. For example, TSF-DelRetGen requires training a classifier to locate style-carrying spans of text. STRAP fine-tunes a GPT-2 model to learn how to paraphrase. IMaT trains a plain Seq2Seq model from scratch as an LM (not pre-trained). All these methods show limited performance in data hungry cases, partially because the compromised external modules would propagate errors to following parts as they are coupled. LaMer instead, built upon pre-trained LMs, requires neither extra data or additional model training, and thus shows superior performance in few-shot cases. Of note, more unsupervised data will benefit LaMer, but we claim LaMer can do at least as good as many SotA methods in few-shot scenarios. We also include a zero-shot baseline (GPT-3) for reference. Please see revised Section 3.2 for more details about these new experiments.
>
> - **Additional human evaluation.** Thanks for your suggestions. We have added human evaluation results to show how well SAS-aligned pairs are similar in content in newly added Table 5.
>
> - **Which version of BLEU?** We use BLEU-*ref* on multiple references (for sentiment and formality TST). We have added this detail to the paper (see Section 3.2 and footnote 5).
>
> - **Gap between in-house and officially reported results.** We have run their models several times and still cannot reach their reported results. We find we can either approach their reported ACC or BLEU but not at the same time. Similar observations can be found [here](https://github.com/luofuli/DualRL/issues/15), and [one published paper](https://aclanthology.org/2020.inlg-1.25.pdf) (see Table 2, Dual RL, h-BLEU = 17.71, which is BLEU-*ref*) [1]. Finally, we pick the combination with relatively higher acc and lower BLEU (which also leads to the highest GM). We are happy to update this result if you have further suggestions!
>
> - **Why RD has good performance (in ACC, and sometimes in BLEU)?** RD means the source and target sentences are randomly mapped, which means if we train a text2text LM on such data, it can still be aware of opposite styles (good ACC) but has weak supervision to learn how to maintain style-independent content. Our results actually echo this hypothesis: In sentiment and formality TST, LM achieves much higher BLEU than RD. For political stance TST, we find mere LM alignment is not powerful enough---certain entities that appear in scene graphs seem to be very important---LM + SAS brings the most benefit in this case. The reason why RD sometimes can still lead to good BLEU could be the unsupervised data may have many duplicated sentences already, which is especially true in sentiment TST. Please take a look at Table 7 in the appendix: We analyze the SAS before (RD) and after SAS alignment, and compare it with that of human references. In sentiment TST, even though we do not align the data, the SAS of RD (0.55) already approaches that of human references (0.73), which are already pairs in parallel. Training a text2text LM on such "already weakly parallel" (though because of duplication) data, and with our IL refinement (upweighting really good parallel but penalizing bad ones) could probably lead to good BLEU. It becomes chanllenging in political stance TST, which RD has inferior BLEU and overall performance.
>
> - **SAS is helpful for content preservation?** In our revised Table 1 we have used GM to measure the overall performance of TST models. Given this new setting, we have updated some records in the table to show the best GM combinations. In this new setting, SAS shows an obvious boost on BLEU. Actually in our original paper, SAS will lead to BLEU increase in two out of three tasks (formality and political, which are the more challenging TST tasks).
>
> - **A new baseline.** Thanks for your suggestion. We have included this work as one of our baselines. Please take a look at revised Table 1, and newly added Tables 2, 3.
>
> Finally, we thank the reviewer for the thoughtful comments and constructive suggestions!
>
> [1] [Stable Style Transformer: Delete and Generate Approach with Encoder-Decoder for Text Style Transfer](https://aclanthology.org/2020.inlg-1.25.pdf)

---

> > ### Comment · Reviewer_53GA · 2021-11-25
> > **Raised my point and feedbacks to author response**
> >
> > Thank you for your explanations and added experimental results. They look good. I still two comments based on my previous points:
> > 1. My first comment in my last review is actually not exactly about low-resource. What I am concerned is that what if there are no or very low percentage of pairs that share similar content in the data distribution? I am wondering whether the authors can add more results on what is the percentage of high-quality pseudo pairs that can be found for each dataset via human evaluation. Especially for the formality dataset, this dataset is actually comprised of parallel pairs so it is very easy to find out pseudo pairs using some unsupervised sentence matching models, however, what if we split the dataset and only use the first half from the formal style and the second half from the informal style? In this case, there will be no pairs existing in the data, and then could we still find out pseudo pairs of good enough quality to make this method effective?
> > Overall, my concern of this method is still whether its effectiveness depends on whether there exists high-quality pairs in the original data distribution. If there is very low percentage of high-quality pseudo pairs, would this method fail? If you can resolve this puzzle, I can further raise my point.
> > 2. In terms of the previous work results replication, I once ran the source code of DualRL and roughly remember that I can obtain the results shown in the paper. So I am not sure what is the reason why you cannot replicate the results in that paper. Have you contacted the original authors for the replication issue?

---

> > > ### Author Response · Authors · 2021-11-27
> > > **Thank you and further clarifications**
> > >
> > > Thank you for clarifying your questions and for updating your score!
> > >
> > > **About Question 1**
> > >
> > > In our few-shot experiment, we actually try to get at what you are suggesting here. We sample the formal and informal sentences separately, not in pairs. This way we are very likely to be left with formal and informal sentences that have different content distribution (since they are shuffled out of sync). Specifically, the process of our few-shot experiments is as follows:
> > >
> > > 1. We sample 1% of the data points from each of the formal and informal sentences (which corresponds to around 520 formal and 520 informal sentences). Note that we sample the formal and informal sentences separately, meaning that we do not sample formal-informal pairs. Now we have the new $D^{\textrm{src}}$ (formal) and $D^{\textrm{tgt}}$ (informal); each with a size of 520.
> > > 2. We run alignment on $D^{\textrm{src}}$ and $D^{\textrm{tgt}}$. Each $s^{\textrm{src}} \in D^{\textrm{src}}$ will be mapped to multiple $s^{\textrm{tgt}} \in D^{\textrm{tgt}}$ by the alignment algorithm. The number of these one-to-N pairs is 520 since we have 520 $s^{\textrm{src}} \in D^{\textrm{src}}$.
> > > 3. We then split these one-to-N pairs into one-to-one pairs, resulting in 1800 $s^{\textrm{src}} - s^{\textrm{tgt}}$ pairs.
> > > 4. We use the one-to-one pairs for MLE training, and the one-to-N pairs for IL refinement.
> > >
> > > We measure the SAS score of our few-shot dataset before and after we perform alignment. The scores are 0.01 (before alignment, i.e., random alignment), 0.03 (after LM alignment), 0.09 (after LM + SAS alignment). Compared with 0.29 SAS for LM + SAS alignment in full dataset (shown in Table 7), it may partially explain why we see a 40 point $J$-score drop (absolute) when the data is decreased to 1% (see Table 2 in our revised version). However, other baselines show even worse performance as their external modules require training on ample data to work properly. In general, if there is very low percentage of high-quality pseudo pairs (such as 1% few-shot cases), LaMer has relatively robust performance compared to other strong baselines (at least no worse than them), and outperforms them when the full data is available.
> > >
> > > We agree that additional human evaluations to measure the percentage of high-quality pseudo pairs will be an informative addition to the paper. We started running this experiment this morning and though it is somewhat time-consuming, we are confident the experiments can be done in the next week, allowing us to add these results to our camera ready. However, note that absent these experiments, the human reference section of Table 7 actually shows the how much parallel "human alignment" can achieve: Since the test sets are written by human (i.e., Human References in Table 7) and they have to be in parallel, the SAS score calculated over these human references actually demonstrates what’s the upper bound of the parallelism shown in human demonstrations. Table 7 shows that LaMer’s SAS alignment manages to reach human-level parallelism, and as discussed in the last paragraph, we see improvements in few-shot cases as well.
> > >
> > > For the TST tasks that source and target style use nearly different forms of language (e.g., ancient to modern English TST), we agree LaMer’s performance could be limited if we still choose scene graph alignment (i.e., SAS alignment) since there are little similarities in surface level among entities. We thus recommend using the (multilingual) LM alignment (our second proposed alignment method) to align the data as an alternative. We have discussed this in the revised section 2.2.
> > >
> > > **About Question 2**
> > >
> > > Thank you for the suggestion. We have indeed contacted the authors of DualRL, and we are looking forward to their response. However, to err on the side of caution, we plan to report their official results (and maybe add our results to appendix) if they don’t get back to us. We will reflect our edits in camera ready version if we have any further updates.
> > >
> > > Hope our answers can clarify your concerns. We appreciate your kind understanding and professional suggestions!

---

### Author Response · Authors · 2021-11-17
**General Response to All Reviewers**

We thank all the reviewers for their valuable comments and constructive feedback. We are glad that reviewers found strengths in our paper’s novelty, experimental procedure, and contribution to the community. Different from existing methods, LaMer uses scene graphs to align sentences from different styles, requiring neither extra data nor additional systems training.

We have revised the paper according to the suggestions (highlighted in bronze in the paper). We summarize the highlights from the revision below and address each reviewer’s feedback separately as well.

- **The performance of LaMer in few-shot cases (R1, R5).** To study the performance of LaMer in extreme data hungry cases, we run new experiments in few-shot scenarios (where only 1% of training data is available). We compare LaMer with several LM-based baselines and the GPT-3 based zero-shot TST method. We present the results in Table 2, and discuss the main differences between LaMer and other methods in Table 3. See our revised evaluation part (section 3.2) for the few-shot experiments.

- **The evidence that there is self-parallelism within unsupervised data (R4, R5).** We justify our assumption regarding self-parallelism in certain unsupervised datasets through empirical observation on the data: In Table 7 (located in the appendix) we have shown that if NO SAS alignment happens (RD), there would be huge discrepancy on the average SAS score between training data and human references (e.g., in Formality TST, 0.29 with aligned by SAS and 0.01 without alignment). Since the human references are already parallel style transfer sentences, our success on the alignment demonstrates that there is self-parallelism existing in the unsupervised data. Additional human evaluation confirms that our aligned sentences are indeed similar in style-independent content---please take a look at Table 5 and related discussion.

- **Additional Human judgement about scene graph alignment (R1, R2, R3).** We have added the additional human evaluation results about how well the scene graph based alignment works based on human judgement in Table 5. We also added new statistics about human ratings in Table 6.

- **Qualitative demonstration of scene graph entities, procedure of the parsing, and the scene graph parser we used (R2, R3, R4).**
 We have added a visualization of scene graph alignment (Figure 2), described the parsing procedure (revised section 2.2), and noted the parser that we used (footnote 1).

- **New baseline (R1).** IMaT was added as an additional baseline; We have included its results in Tables 1, 2, and 3.

- **Better evaluation metrics (R3, R4, R5).** Recent TST have pointed out flaws of commonly used evaluation metrics (such as BELU). We evaluate LaMer’s performance with the more advanced $J$-score framework recently proposed by [1] in Table 2, while we still keep the ACC and BLEU in Table 1 for easy comparison with other baselines. We also added GM (geometric mean of ACC and BLEU) as an overall metric to better present our results (shown in revised Table 1).

- **Scope of this work (R4, R5).** We have added additional discussion in section 2.2 regarding the scope of our SAS alignment and describe how other non-attribute-based alignment methods proposed by LaMer (such as multilingual LM alignment) can be used for TST scenarios where entities may not be shared across styles (e.g., multilingual TST, or ancient to modern English TST).

- **Minor fix on presentations (Thanks all reviewers!).** We have fixed the links to the appendix and fixed several typos.

We hope our revised paper and additional experiments can provide further clarity. Please let us know if any improvement can be made!


[1] [Reformulating Unsupervised Style Transfer as Paraphrase Generation](https://arxiv.org/pdf/2010.05700.pdf)

---

### Decision · Program_Chairs · 2022-01-20

**Decision:**

Accept (Poster)

**Comment:**

The paper proposes a new method for unsupervised text style transfer by assuming there exist some pseudo-parallal sentences pairs in the data. The method thus first mines and constructs a synthetic parallel corpus with certain similarity metrics, and then trains the model via imitation learning. Reviewers have found the method is sound and the empiricial results are decent. The assumption on pseudo-parallal pairs would limited the application of the methods in other settings where the source/target text distributions are very different. The authors have added discussion on this limitation during rebuttal.